# Mitigating Surgical Data Imbalance with Dual-Prediction Video Diffusion Model

Danush Kumar Venkatesh [1][*]   Adam Schmidt [2]   Muhammad Abdullah Jamal [2]   Omid Mohareri [2]

## Abstract

Surgical video datasets are essential for scene understanding, enabling procedural modeling and intra-operative support. However, these datasets are often heavily imbalanced, with rare actions and tools under-represented, which limits the robustness of downstream models. We address this challenge with *SurgiFlowVid*, a sparse and controllable video diffusion framework for generating surgical videos of under-represented classes. Our approach introduces a dual-prediction diffusion module that jointly denoises RGB frames and optical flow, providing temporal inductive biases to improve motion modeling from limited samples. In addition, a sparse visual encoder conditions the generation process on lightweight signals (e.g., sparse segmentation masks or RGB frames), enabling controllability without dense annotations. We validate our approach on three surgical datasets across tasks including action recognition, tool presence detection, and laparoscope motion prediction. Synthetic data generated by our method yields consistent gains of 10–20% over competitive baselines, establishing *SurgiFlowVid* as a promising strategy to mitigate data imbalance and advance surgical video understanding methods.

## 1. Introduction

Robotic-assisted minimally invasive surgery (RAMIS) has become a cornerstone of modern surgical practice, offering patients reduced trauma, faster recovery, and fewer complications (Haidegger et al., 2022; Taylor et al., 2016). However, operating using an endoscopic video feed rather than direct vision introduces challenges such as limited depth perception, reduced haptic feedback, and altered hand–eye coordination. These limitations increase both the cognitive and technical demands placed on surgeons during procedures (Sørensen et al., 2016; Dagnino & Kundrat, 2024).

The emerging field of *Surgical Data Science* seeks to address these challenges by developing computational methods that leverage the video data generated during surgery. In particular, deep learning (DL) methods could be utilized to understand the surgical scene, thereby supporting intra-operative decisions and reducing the burden on surgeons. Surgical video datasets, therefore, play a central role in enabling tasks, including surgical phase and gesture recognition (Padoy et al., 2012; Funke et al., 2025; 2019a), instrument detection and segmentation (Nwoye et al., 2022b; Kolbinger et al., 2023), tool tracking (Schmidt et al., 2024), and skill assessment (Funke et al., 2019b; Hoffmann et al., 2024). However, despite recent efforts to release annotated datasets (Ayobi et al., 2025; Psychogyios et al., 2023; Wang et al., 2022), these resources remain heavily imbalanced, with rare actions, steps, or tool usages under-represented (see Fig. 1). Such skewed distributions limit the generalization of DL models. Common approaches such as class-sampling and augmentation can increase the frequency of these samples but do not contribute to the diversity of the dataset.

The data imbalance challenge in surgical datasets have motivated increasing interest in synthetic data generation. With the advent of diffusion models (DMs) (Ho et al., 2020; Dhariwal & Nichol, 2021a), synthetic surgical images have been successfully utilized to augment real datasets, thereby reducing imbalance and enhancing downstream performance (Venkatesh et al., 2025b; Frisch et al., 2023; Nwoye et al., 2025). However, extending DMs to surgical video generation remains underexplored due to the substantial demands in data and compute. While recent progress in video synthesis is promising, controllability is especially critical in the surgical domain, where specific tools and anatomical structures must appear in procedure-dependent contexts (e.g., laparoscopy vs. robotic surgery). Prior work often relies on dense per-frame segmentation masks to control video generation (Biagini et al., 2025; Sivakumar et al., 2025; Yeganeh et al., 2024; Iliash et al., 2024; Cho et al., 2024), but these require costly expert annotations that are

---

[1]Department of Translational Surgical Oncology, NCT/UCC Dresden, a partnership between DKFZ, Faculty of Medicine and University Hospital Carl Gustav Carus, TUD Dresden, HZDR, Germany. [2]Intuitive Surgical, Inc., California, US. [*] Work done during an internship. Correspondence to: Adam Schmidt <adam.schmidt@intusurg.com>.

*Proceedings of the $43^{rd}$ International Conference on Machine Learning*, Seoul, South Korea. PMLR 306, 2026. Copyright 2026 by the author(s).

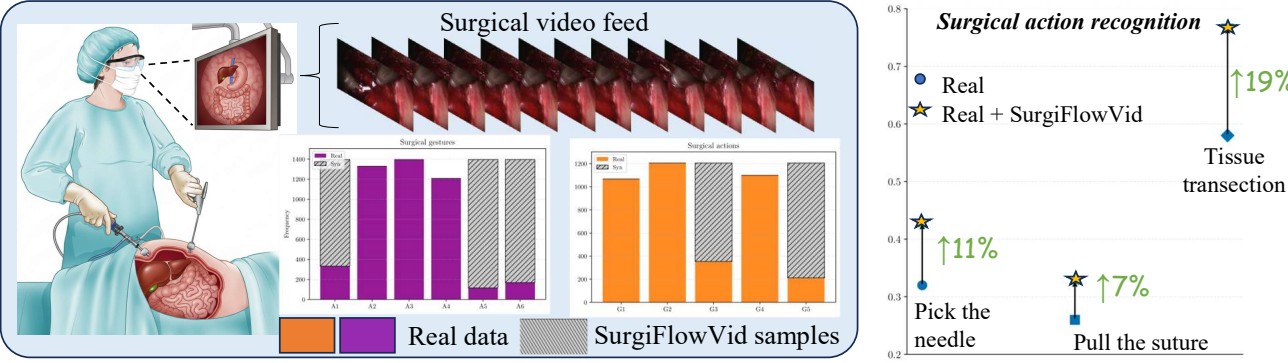

Figure 1. **Data challenge in the surgical domain**. During a laparoscopic procedure, the surgeon operates via the endoscopic video feed (video on the monitor). ML models can leverage these videos for providing guidance through surgical scene understanding. However, the datasets are *skewed* as shown in the bar plots. We aim to mitigate data imbalance with synthetic samples. The right plot shows improvements from adding samples generated from our approach (SurgiFlowVid).

rarely available. In practice, surgical datasets typically contain only sparse (temporal) segmentation masks—or none at all—while under-represented classes are particularly scarce. This raises a critical question: *how can generative models improve learning for under-represented classes when only sparse or no conditional signals are available?*

To address this challenge, we propose *SurgiFlowVid* (**Surgi**cal **Flow**-inducted **Vid**eo generation), a diffusion-based framework designed to synthesize spatially and temporally coherent surgical videos of under-represented classes. We introduce a dual-prediction approach that jointly denoises RGB frames and optical flow maps, providing inductive biases to improve motion modeling from limited data. Beyond text prompts, SurgiFlowVid can condition directly on RGB frames or sparse segmentation masks, when available, via a visual encoder. While video DMs typically rely on heavy compute, our approach is tailored to the constrained settings common in healthcare, ensuring practical applicability.

SurgiFlowVid generates diverse and temporally coherent videos of under-represented classes, which we use to augment real datasets and evaluate performance on short-context, fine-grained surgical tasks. Such tasks capture critical tool–tissue interactions and have been shown to provide meaningful feedback during surgical procedures (Ma et al., 2022). By tackling the challenges of data imbalance, our approach advances robust DL methods for surgical video understanding applications. We summarize our contributions as follows:

1. We address the critical challenge of data imbalance in surgical datasets by synthesizing video samples of under-represented classes with diffusion models, providing a principled way to augment real world datasets.

2. We introduce *SurgiFlowVid*, a surgical video diffusion

framework equipped with a dual-prediction diffusion U-Net that leverages both RGB frames and optical flow to capture spatio-temporal relationships, even in the minimal available video samples of under-represented classes. In addition, a visual encoder enables conditioning on sparse conditional frames when available, removing the need for costly dense annotations.

3. We extensively evaluate the proposed framework on three surgical datasets across tasks like action recognition, tool presence detection and laparoscope motion detection indicating performance gains of 10–20% over strong baselines highlighting the effectiveness of our approach in advancing robust surgical video understanding models.

## 2. Related Work

**Synthetic data in surgery** 2D synthetic laparoscopic surgical images generated using GANs (Goodfellow et al., 2014) and diffusion models (DMs) (Dhariwal & Nichol, 2021b; Sohl-Dickstein et al., 2015) have been shown to enhance downstream tasks (Venkatesh et al., 2024; 2025b; Frisch et al., 2023; Nwoye et al., 2025; Allmendinger et al., 2024; Martyniak et al., 2025; Pfeiffer et al., 2019). However, these approaches remain limited to static image generation and fail to capture the temporal context essential for surgical videos, which are the primary data source in real-time procedures. While diffusion models have also shown success in medical imaging domains such as MRI and CT (Dorjsembe et al., 2022; Khader et al., 2023; Zhao et al., 2026), these modalities differ fundamentally from surgical video data.

**Surgical Video Synthesis** Although laparoscopic video synthesis has attracted increasing attention in recent years, its potential for addressing data imbalance in surgical tasks remains underexplored. Endora (Li et al., 2024) introduced

unconditional video generation, while MedSora (Wang et al., 2024) proposed a framework based on a Mamba diffusion model. However, both approaches lacked controllability, which is crucial for generating task-specific videos that can mitigate data imbalance. Iliash et al. (2024) and Sur-Gen (Cho et al., 2024) extended video generation by conditioning on pre-defined instrument masks and they require vast quantities of labeled real data ($\approx$ 200K videos), which restricts its applicability to well-studied procedures, such as cholecystectomy (Nwoye et al., 2022a; Twinanda et al., 2016), and prevents its generalization to less documented surgeries.

Other works, such as VISAGE (Yeganeh et al., 2024) and SG2VID (Sivakumar et al., 2025), condition generation on action graphs which require curated datasets with detailed annotations and they are often unavailable for many surgical procedures. SurgSora (Chen et al., 2025) conditions generation on user-defined instrument trajectories, whereas Bora (Sun et al., 2024) leverages large language models (LLMs) to generate instruction prompts for controllability. More recently, SurV-Gen (Venkatesh et al., 2025a) was proposed as a framework for generating samples of rare classes and combined with a rejection sampling strategy to filter out degenerate cases (poor consistency) of synthetic videos from a large candidate pool. Although there exists plethora of state-of-the-art video diffusion models for the natural domain (Rombach et al., 2022b; Yang et al., 2025b; Agarwal et al., 2025; Polyak et al., 2024), adapting them for the surgical domain is challenging due to the large amounts of curated video data and compute needed to train them. Additional related work is in the appendix (A).

Our approach, although closely related to SurV-Gen, introduces notable advantages: by incorporating optical flow as an inductive bias, we generate temporally coherent and plausible videos without the need for rejection sampling. Additionally, by conditioning on sparse segmentation masks or RGB frames, we achieve greater controllability and diversity in generating under-represented classes.

## 3. Method

Our goal is to alleviate data imbalance by generating synthetic surgical videos of under-represented classes, a task that is made difficult by the limited data available to model spatial and temporal dynamics accurately. To address this, we introduce *SurgiFlowVid*, which includes a multi-stage conditional training with two different modules as follows,

(i) *Dual-prediction diffusion U-Net:* We introduce a U-Net module that jointly predicts RGB frames and optical flow maps during training, enabling the model to capture temporal motion alongside spatial appearance which cannot be reliably inferred from RGB appearance alone.

(ii) *Sparse conditional guidance:* Dense segmentation masks are rarely available in surgical datasets, and relying solely on text or label conditioning provides weak guidance. Instead, we design a sparse visual guidance encoder that conditions the diffusion process on either the available sparse segmentation masks or the RGB frames from the input video. Our model supports both text-based unconditional generation and conditional generation with sparse masks (if available), generating under-represented class samples with spatio-temporal consistency. The overview of our approach is shown in Fig. 2.

### 3.1. Surgical Video Generation

We build our framework on top of the SurV-Gen model, which follows a two-stage training strategy. In the first stage, Stable Diffusion (SD) (Rombach et al., 2022a) is adopted as the base text-to-image model, where the diffusion process is performed in the latent space. An image $x_0$ is first encoded into $z_0$ via an encoder $E(x_0)$, and during the forward diffusion process $z_0$ is iteratively perturbed as $z_t = \sqrt{\bar{\alpha}_t}\, z_0 + \sqrt{1 - \bar{\alpha}_t}\, \epsilon$, with $\epsilon \sim \mathcal{N}(0, I)$, $\alpha_t = 1 - \beta_t$, and $\bar{\alpha}_t = \prod_{s=0}^{t} \alpha_s$, where $\beta_t$ determines the noise strength. A denoising network $\epsilon_\theta(\cdot)$ is trained to reverse this process by minimizing the reconstruction loss

$$\mathcal{L} = \mathbb{E}_{E(x_0),y,\epsilon,t}\big[\|\epsilon - \epsilon_\theta(z_t, t, y)\|_2^2\big], \qquad (1)$$

where $y$ denotes the text prompt associated with $x_0$. In the second stage, the fine-tuned spatial layers of the SD model is extended to operate directly on surgical video sequences. Temporal transformer blocks (Vaswani et al., 2017) are inserted after each spatial block while keeping spatial layers frozen, thereby focusing the training on temporal dynamics. Given a video tensor $v \in \mathbb{R}^{b \times c \times f \times h \times w}$, where $b$ is the batch size, $f$ the number of frames, $h$, $w$ and $c$ are the height, width and channel dimensions respectively, the temporal layers reshape $v$ to $(bhw) \times f \times c$ and apply self-attention: $v_{\text{out}} = \text{Softmax}\left(\frac{QK^T}{\sqrt{c}}\right) V$ with $Q = v_{\text{in}} W_Q$, $K = v_{\text{in}} W_K$, and $V = v_{\text{in}} W_V$ as the query, key, and value projections. Cross-frame attention captures motion dynamics, but relying solely on it or on text and label conditioning is insufficient to model tool and tissue motion.

### 3.2. Dual-prediction module

In our approach, we modify the U-net such that optical flow, $p$, is taken as an input along with input tensor, $v$. Given two consecutive frames $v_1, v_2 \in \mathbb{R}^{3 \times H \times W}$, the optical flow is computed as $D_t(v_1, v_2) = (d_1, d_2)$, which encodes the pixel displacement at location $(v_1, v_2)$. We convert $D_t$ into an RGB image by computing a normalized magnitude

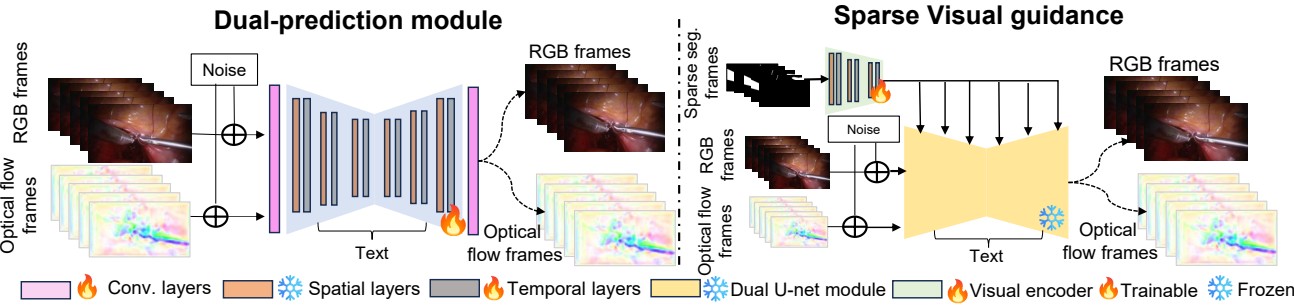

*Figure 2.* **SurgiFlowVid approach.**The dual-prediction diffusion U-Net module reconstructs both RGB and optical flow frames from noised inputs to capture spatio-temporal dynamics from limited data. Sparse visual encoder is trained with segmentation masks (if available) or RGB frames for conditional generation; optical flow is used only during training.

$r(v_1, v_2)$ and angle $\theta$:

$$r(v_1, v_2) = \frac{\sqrt{\widehat{d}_1^2 + \widehat{d}_2^2}}{\|D_t\|_{\max} + \varepsilon}, \; \theta(v_1, v_2) = \frac{1}{\pi} \operatorname{atan2}(\widehat{d}_2, \widehat{d}_1)$$

where $\widehat{d}_1, \widehat{d}_2$ denote the normalized flow components and $\varepsilon > 0$ ensures numerical stability. The angle $\theta$ is mapped to a color, while the magnitude $r$ attenuates this color to produce the RGB encoding resulting in the flow tensor $p^{c \times (f-1) \times h \times w}$. We define the *dual-prediction* diffusion U-Net by modifying its input and output layers to process RGB frames and optical flow jointly. Specifically, the first layer is adapted to accommodate both tensors, $v$ and $p$, while the final layer is modified to predict both RGB and flow frames. These layers are trained together with the temporal attention layers using the loss function ($L$) defined as,

$$\mathcal{L} = \mathbb{E}_{E(x_0), y, \epsilon, t} \left[ \|\epsilon - \epsilon_\theta(z_t, t, y)\|_2^2 + \lambda_p \|\epsilon - \epsilon_\theta(z_p, t, y)\|_2^2 \right] \tag{2}$$

where $z_p$ is the noised optical flow frames and $\lambda_p$ is a weighting parameter. The model jointly denoises each chunk of RGB and flow frames. We freeze the spatial layers in this stage and optical flow is used solely during training.

### 3.3. Sparse visual guidance

To incorporate conditional guidance, we extend the sparse condition encoder proposed in SparseCtrl (Guo et al., 2024), which propagates sparse signals (e.g., frames) across time using spatial and temporal layers to improve consistency between conditioned and unconditioned frames. In our framework, we integrate the dual-prediction U-net and re-define the *sparse visual encoder* as a lightweight module that encodes only the sparse conditional frames. The U-Net backbone is frozen, and only the visual encoder (SVE) is optimized using the loss in Eq. 2. By incorporating optical flow into the loss, we explicitly supervise both motion and structure, allowing the model to move beyond appearance

propagation alone, thereby reducing data requirements and improving robustness. Formally, given sparse conditional signals $s_s \in \mathbb{R}^{3 \times H \times W}$ (e.g., RGB frame or segmentation mask) and a binary mask $m \in \{0, 1\}^{1 \times H \times W}$ indicating whether a frame is conditioned, the sparse encoder input is constructed as $\hat{c} = [\, s_s \,\|\, m \,]$ where $\|$ denotes channel-wise concatenation. This design offers flexibility by enabling diverse conditional inputs to guide the generation process. At inference, sparse frames from the real dataset are selected and injected into specific temporal indices to condition the generation process.

## 4. Experiments

In this section, we outline our experimental setup, evaluation schemes and the downstream tasks we evaluate the generated synthetic datasets. Particularly, we focus on fine-grained surgical tasks of step and tool recognition and laparoscope movement detection. We define an under-represented class at the clip level as one with a high imbalance ratio (refer to App. C for details).

### 4.1. Datasets

(i) **SAR-RARP50** consists of robotic radical prostatectomy videos from 50 patients, with a split of 35, 5, and 10 patients for training, validation, and test sets, respectively (Psych-ogyios et al., 2023). The annotated surgical actions include: picking up the needle (A1), positioning the needle (A2), pushing the needle through tissue (A3), pulling the needle (A4), cutting the suture (A5), tying the knot (A6), and returning the needle (A7). Since action A6 occurs only once in the test set, it is omitted from evaluation. The under-represented classes in this dataset are A1, A5, and A7. The primary task involves recognizing the surgical action at time $t$ given a video clip. In addition, segmentation masks are available for nine classes collected at 1fps. Using these masks, we construct the task of surgical tool presence detection, where the objective is to identify which instruments are present in

a given surgical video.

(ii) **GraSP** includes robotic prostatectomy procedures (Ayobi et al., 2025). It consists of 13 patients with a two-fold cross-validation setup, where five patients are held out for testing. The dataset contains annotations for 20 different surgical actions. For this study, we focus on a subset of five actions: pulling the suture (G1), tying the suture (G2), cutting the suture (G3), cutting between the prostate and bladder neck (G4), and identifying the iliac artery (G5). All classes except G5 are under-represented.

(iii) **AutoLaparo** contains laparoscopic hysterectomy videos from 21 patients, with annotations describing the movements of the laparoscope (Wang et al., 2022). In total, it contains approximately 300 clips, each lasting 10 seconds, covering six motion types: up, down, left, right, zoom-in, and zoom-out. The laparoscope motion occurs precisely at the 5th second of each clip, enabling the formulation of two tasks. In the *online* recognition setting, only the first 5 seconds are provided to the model to predict the upcoming motion, which is particularly relevant for real-time applications. In the *offline* setting, the entire clip is available, and the task is to classify the laparoscope motion using full temporal context. These annotations can be used for developing automatic field-of-view control systems. We considered all movement classes to be under-represented.

### 4.2. Baselines

(i) *Generation*: We evaluate our method against recent surgical video diffusion models. Endora (Li et al., 2024) is a fully transformer-based unconditional diffusion model, which we train separately on each minor class due to its lack of controllability. SurV-Gen (Venkatesh et al., 2025a) serves as a conditional baseline with both text and label guidance. We also include its rejection sampling (RS) strategy, which filters out degenerate generations and thus represents a strong reference baseline. In addition, we adapt the SparseCtrl (Guo et al., 2024) model, an effective conditional video diffusion approach that generates videos conditioned on text and sparse conditional masks. The SurV-Gen model acts as the ablation of our approach without the dual-prediction module ($\lambda_p$=0). Additionally, we ablate the model without the SVE module. We maintain a patient specific data split and generate four second videos of 16-frames aligning with the requirements of the downstream task.

(ii) *Downstream task*: For surgical action (step) recognition, we report the averaged video-wise Jaccard index per class and mean average precision (mAP) averaged across videos for the SAR-RARP50 and GraSP dataset respectively following (Psychogyios et al., 2023; Ayobi et al., 2025). For the multi-label classification of surgical tools we measure the Dice score and report F1 for laparoscope motion detection. We apply inverse frequency balancing with video

frame augmentation only on the real datasets and downsample the synthetic videos to the same resolution for fair comparison. Especially, we add synthetic videos of under-represented classes to the real dataset and leave the well balanced classes undisturbed. Please refer to the appendix for details on model training(D.4) and additional experiments and evaluations(B). Together, these baselines span unconditional, conditional, and sparse conditional video diffusion approaches, providing a comprehensive reference for evaluating our method.

### 4.3. Evaluation scheme

We systematically structure our experimental design into three parts to evaluate the role of synthetic data in addressing class imbalance.

(i) **Synthetic data attributes:** We analyze which attributes of synthetic data are essential for improving downstream performance. To this end, we conduct controlled experiments on the surgical action recognition task. First, we *duplicate* the training set and train for the same number of epochs to evaluate whether performance gains arise from true data diversity rather than simple repetition. Second, to assess the effects of *spatial* and *temporal* consistency, we simulate degraded data by applying elastic deformations and noise to video frames (disrupting spatial structure) and by shuffling frames (disrupting temporal order). Third, we evaluate the effect of *sparse conditioning* by constructing videos from only sparse frames and examining their impact on downstream performance.

(ii) **Class modeling:** We investigate whether synthetic data is more effective when all under-represented classes are modeled jointly or when each class is modeled separately.

(iii) **Downstream tasks:** We evaluate the effect of synthetic data on three surgical downstream applications: surgical action recognition, surgical tool presence detection, and laparoscope motion prediction.

*Table 1.* **Attributes of synthetic** data experiment on the under-represented classes of the SAR-RARP50 dataset.

| Method | A1 | A5 | A7 |
|---|---|---|---|
| Real | $0.32_{\pm 0.19}$ | $0.10_{\pm 0.04}$ | $0.32_{\pm 0.15}$ |
| Data duplicate | $0.32_{\pm 0.17}$ | $0.11_{\pm 0.02}$ | $0.32_{\pm 0.13}$ |
| Frame shuffle | $0.30_{\pm 0.14}$ | $0.06_{\pm 0.09}$ | $0.30_{\pm 0.17}$ |
| Sparse frame | $0.28_{\pm 0.14}$ | $0.05_{\pm 0.05}$ | $0.29_{\pm 0.10}$ |
| Noisy frame | $0.29_{\pm 0.14}$ | $0.04_{\pm 0.05}$ | $0.29_{\pm 0.10}$ |

## 5. Results & Discussion

**Synthetic data attributes** Our evaluation of different synthetic data attributes for under-represented classes in the SAR-RARP50 dataset is in Tab. 1. Merely duplicating the

*Table 2.* **Surgical action recognition on the SAR-RARP50 dataset**. Under-represented classes are highlighted, and Jaccard index is reported. *Ic* denotes individual class modeling, and RS indicates rejection sampling. †denotes the ablation models. Addition of synthetic samples from SurgiFlowVid indicates comprehensive gains for the under-represented classes.

| Training data | Cond. type | | Pick the needle | Position the needle | Push the needle | Pull the needle | Cut the suture | Return the needle | Mean. |
|---|---|---|---|---|---|---|---|---|---|
| | Text | Sparse mask | | | | | | | |
| Real | – | – | $0.32_{\pm0.19}$ | $0.66_{\pm0.09}$ | $0.78_{\pm0.10}$ | $0.61_{\pm0.09}$ | $0.10_{\pm0.04}$ | $0.32_{\pm0.15}$ | $0.46_{\pm0.08}$ |
| Real + Endora | – | – | $0.32_{\pm0.14}$ | $0.63_{\pm0.05}$ | $0.76_{\pm0.07}$ | $0.61_{\pm0.11}$ | $0.08_{\pm0.04}$ | $0.33_{\pm0.10}$ | $0.45_{\pm0.05}$ |
| Real + SurV-Gen (w/o RS) † | – | – | $0.31_{\pm0.19}$ | $0.64_{\pm0.07}$ | $0.77_{\pm0.06}$ | $0.60_{\pm0.10}$ | $0.13_{\pm0.10}$ | $0.37_{\pm0.18}$ | $0.46_{\pm0.03}$ |
| Real + SurV-Gen (RS) | – | – | $0.35_{\pm0.12}$ | $0.63_{\pm0.02}$ | $0.77_{\pm0.03}$ | $0.61_{\pm0.08}$ | $0.14_{\pm0.09}$ | $0.39_{\pm0.15}$ | $0.48_{\pm0.06}$ |
| Real + SparseCtrl | | RGB | $0.36_{\pm0.17}$ | $0.65_{\pm0.06}$ | $0.78_{\pm0.07}$ | $0.64_{\pm0.11}$ | $0.09_{\pm0.07}$ | $0.40_{\pm0.12}$ | $0.48_{\pm0.04}$ |
| Real + SparseCtrl | | Seg. | $0.36_{\pm0.14}$ | $0.61_{\pm0.12}$ | $0.77_{\pm0.07}$ | $0.63_{\pm0.11}$ | $0.16_{\pm0.11}$ | $0.38_{\pm0.17}$ | $0.49_{\pm0.04}$ |
| Real + SurgFlowVid † | – | – | $0.43_{\pm0.12}$ | $0.65_{\pm0.07}$ | $0.77_{\pm0.07}$ | $0.63_{\pm0.11}$ | $0.11_{\pm0.03}$ | $0.35_{\pm0.12}$ | $0.49_{\pm0.04}$ |
| Real + SurgFlowvid | | RGB | $0.36_{\pm0.17}$ | $\mathbf{0.67_{\pm0.06}}$ | $0.78_{\pm0.08}$ | $\mathbf{0.65_{\pm0.12}}$ | $0.17_{\pm0.10}$ | $0.42_{\pm0.12}$ | $0.51_{\pm0.04}$ |
| Real + SurgFlowVid | | Seg. | $\mathbf{0.44_{\pm0.18}}$ | $0.66_{\pm0.07}$ | $\mathbf{0.79_{\pm0.08}}$ | $0.64_{\pm0.04}$ | $0.18_{\pm0.09}$ | $0.42_{\pm0.15}$ | $0.52_{\pm0.04}$ |
| Real + SurgFlowVid (*Ic*) | – | – | $0.37_{\pm0.16}$ | $0.65_{\pm0.04}$ | $0.77_{\pm0.07}$ | $0.61_{\pm0.10}$ | $0.14_{\pm0.03}$ | $0.42_{\pm0.18}$ | $0.49_{\pm0.06}$ |
| Real + SurgFlowvid (*Ic*) | | RGB | $0.36_{\pm0.14}$ | $0.65_{\pm0.03}$ | $\mathbf{0.79_{\pm0.15}}$ | $0.64_{\pm0.08}$ | $\mathbf{0.20_{\pm0.09}}$ | $\mathbf{0.52_{\pm0.12}}$ | $\mathbf{0.53_{\pm0.02}}$ |
| Real + SurgFlowVid (*Ic*) | | Seg. | $0.41_{\pm0.19}$ | $0.63_{\pm0.06}$ | $0.77_{\pm0.03}$ | $0.62_{\pm0.12}$ | $0.10_{\pm0.05}$ | $0.38_{\pm0.16}$ | $0.48_{\pm0.06}$ |

training set does not improve performance, as it fails to introduce additional sample diversity. Frame shuffling causes a slight decline in performance, underscoring the importance of temporal consistency in video-based tasks. Similarly, injecting noise into frames or conditioning only on sparse frames results in a more substantial drop of about 3–5%. Together, these findings reveal three key aspects: (i) synthetic data must not simply replicate the training set, but rather provide *data diversity*, (ii) maintaining *temporal consistency* is critical, and (iii) preserving *spatial structure* is essential to sustain downstream performance. Overall, this analysis underlines that downstream tasks inherently require both spatial and temporal consistency, and synthetic data must therefore satisfy both to be effective. Additional results from other datasets are in appendix (Sec. B.1).

**Surgical action recognition.** (i) *SAR-RARP50*: The results of surgical action recognition task is reported in Tab. 2. The SurV-Gen model with rejection sampling achieves better performance on under-represented classes compared to using synthetic samples directly, suggesting that its gains stem primarily from the sampling strategy rather than the generative model itself. Synthetic samples from SparseCtrl improves scores across all three underrepresented classes. Our approach, *SurgiFlowVid*, even with text-only conditioning, yields performance improvements in two out of the three under-represented classes, with gains in the range of 3–11%. Adding conditional masks further enhances performance across all classes, with SurgiFlowVid conditioned on segmentation masks achieving improvements of 12%, 8%, and 10% were noticed with for the underrepresented classes. Performance gains are also observed in well-balanced classes, which we attribute to the mutual dependencies among actions. For instance, augmenting data for the "picking the needle" class may indirectly ben-

efit "positioning the needle" class, as the latter can often follow in the surgical workflow. Another noteworthy observation is that modeling each class individually produces a substantial improvement in mean performance, reaching 0.53 compared to 0.46 with real data alone. Particularly notable is the nearly 20% gain for A7, obtained with synthetic samples from SurgiFlowVid (RGB-frame) combined with individual-class training.

(ii) *GraSP*: The qualitative comparison between models is shown in Fig. 3. As seen from Tab. 3 incorporating samples from SurV-Gen yields small performance gains, whereas adding data generated by Endora or SparseCtrl with RGB-frame conditioning results in a decline in mAP score. By contrast, our method, SurgiFlowVid, achieves improvements in two of the four underrepresented classes when used with text only conditioning. Furthermore, with sparse segmentation masks SurgiFlowVid achieves performance gains across all under-represented classes. These results highlight the combined value of our dual-prediction and sparse encoder modules, which enables the model to learn spatio-temporal relationships from limited data more effectively.

**Surgical tool presence detection** The results of the surgical tool presence detection task on the SAR-RARP50 dataset is shown in Tab. 4. Overall, the addition of synthetic samples from generative models leads to consistent performance improvements. This trend can be explained by the fact that the generated surgical videos naturally increase the occurrence of individual tools within the training set. On SAR-RARP50, our approach, SurgiFlowVid, achieves a 10-point improvement over using only real data, compared to a 5-point gain from SparseCtrl. Notably, SparseCtrl's reliance on sparse conditioning yields only limited benefits, improv-

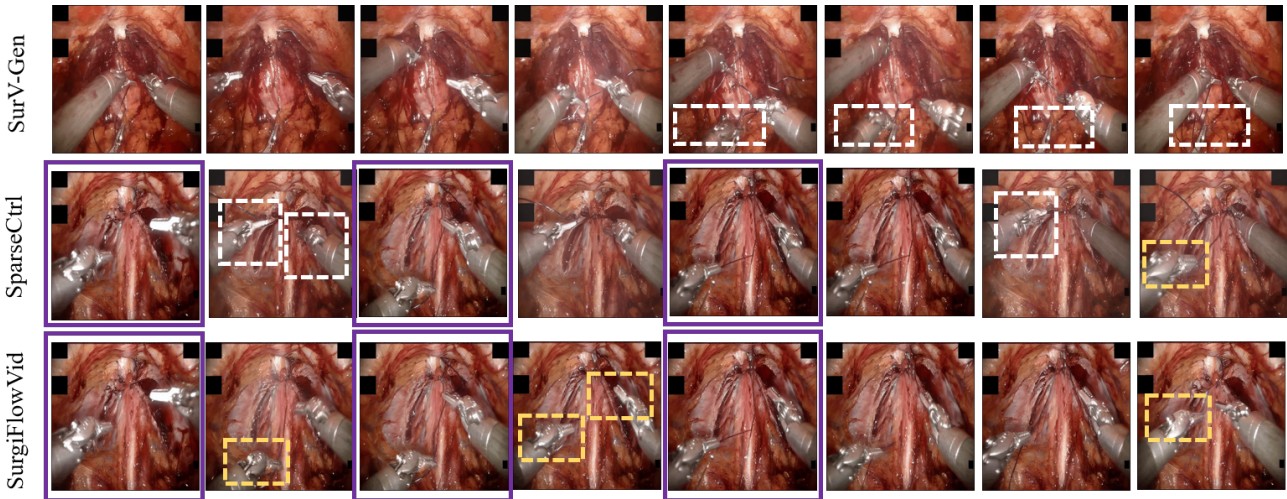

*Figure 3.* **Qualitative results** of the action "tie the suture." Purple boxes denote the sparse RGB conditioning frames. Spurious tools are generated in SurV-Gen (white box, row 1), while SparseCtrl alters tool types compared to the conditioning frames (white box, row 2), reflecting limited spatial consistency. SurgiFlowVid indicates consistent tools across generated frames (yellow boxes, row 3).

*Table 3.* **Surgical step recognition on the GraSP dataset**. The best scores are in **bold** and the mAP scores are reported. Considerable performance gains are noticed for our approach with the sparse RGB frames in comparison to solely using the real dataset.

| Training data | Cond. type | | Pull the suture | Tie the suture | Cut the suture | Cut btw. the prostate | Identify the iliac artery | Mean. |
|---|---|---|---|---|---|---|---|---|
| | Text | Sparse mask | | | | | | |
| Real | – | – | $0.26_{\pm 0.03}$ | $0.44_{\pm 0.01}$ | $0.43_{\pm 0.06}$ | $0.72_{\pm 0.07}$ | $0.52_{\pm 0.08}$ | $0.47_{\pm 0.03}$ |
| Real + Endora | – | – | $0.26_{\pm 0.02}$ | $0.39_{\pm 0.02}$ | $0.40_{\pm 0.05}$ | $0.70_{\pm 0.01}$ | $0.51_{\pm 0.03}$ | $0.45_{\pm 0.04}$ |
| Real + SurV-Gen (w/o RS) † | – | – | $0.30_{\pm 0.01}$ | $0.43_{\pm 0.02}$ | $0.41_{\pm 0.09}$ | $0.71_{\pm 0.04}$ | $0.57_{\pm 0.07}$ | $0.48_{\pm 0.03}$ |
| Real + SurV-Gen (RS) | – | – | $0.30_{\pm 0.02}$ | $0.44_{\pm 0.03}$ | $0.42_{\pm 0.09}$ | $0.73_{\pm 0.02}$ | $0.58_{\pm 0.04}$ | $0.49_{\pm 0.02}$ |
| Real + SparseCtrl | | RGB | $0.27_{\pm 0.01}$ | $0.43_{\pm 0.01}$ | $0.40_{\pm 0.09}$ | $0.71_{\pm 0.04}$ | $0.55_{\pm 0.04}$ | $0.46_{\pm 0.04}$ |
| Real + SurgFlowVid † | – | – | $0.30_{\pm 0.01}$ | $0.43_{\pm 0.03}$ | $0.44_{\pm 0.09}$ | $0.69_{\pm 0.04}$ | $0.60_{\pm 0.07}$ | $0.49_{\pm 0.04}$ |
| Real + SurgFlowvid | | RGB | $\mathbf{0.33}_{\pm 0.01}$ | $\mathbf{0.48}_{\pm 0.02}$ | $\mathbf{0.47}_{\pm 0.01}$ | $\mathbf{0.74}_{\pm 0.02}$ | $0.60_{\pm 0.05}$ | $\mathbf{0.52}_{\pm 0.04}$ |
| Real + SurgFlowVid (*Ic*) | – | – | $0.31_{\pm 0.04}$ | $0.41_{\pm 0.03}$ | $0.42_{\pm 0.04}$ | $0.72_{\pm 0.04}$ | $\mathbf{0.61}_{\pm 0.05}$ | $0.49_{\pm 0.03}$ |
| Real + SurgFlowvid (*Ic*) | | RGB | $0.31_{\pm 0.01}$ | $0.45_{\pm 0.01}$ | $0.43_{\pm 0.03}$ | $0.72_{\pm 0.02}$ | $0.55_{\pm 0.05}$ | $0.50_{\pm 0.02}$ |

*Table 4.* **Surgical tool presence detection on SAR-RARP50 dataset**. Our approach with seg. conditioning outperforms the baseline across all seven tool categories.

| Training data | Tool clasper | Tool wrist | Tool shaft | Suturing needle | Thread | Suction tool | Needle holder | Clamps | Catheter | Mean |
|---|---|---|---|---|---|---|---|---|---|---|
| Real | $0.85_{\pm 0.10}$ | $0.84_{\pm 0.09}$ | $0.88_{\pm 0.07}$ | $0.70_{\pm 0.15}$ | $0.75_{\pm 0.12}$ | $0.69_{\pm 0.11}$ | $0.66_{\pm 0.07}$ | $0.44_{\pm 0.11}$ | $0.46_{\pm 0.08}$ | $0.69_{\pm 0.06}$ |
| Real + SparseCtrl(Seg) | $0.87_{\pm 0.11}$ | $0.83_{\pm 0.05}$ | $\mathbf{0.89}_{\pm 0.06}$ | $0.73_{\pm 0.12}$ | $0.80_{\pm 0.13}$ | $\mathbf{0.79}_{\pm 0.10}$ | $0.74_{\pm 0.09}$ | $0.69_{\pm 0.08}$ | $0.50_{\pm 0.12}$ | $0.74_{\pm 0.03}$ |
| Real + SurgFlowVid(Seg) | $\mathbf{0.88}_{\pm 0.09}$ | $\mathbf{0.85}_{\pm 0.07}$ | $0.88_{\pm 0.10}$ | $\mathbf{0.75}_{\pm 0.11}$ | $\mathbf{0.81}_{\pm 0.09}$ | $0.78_{\pm 0.15}$ | $\mathbf{0.75}_{\pm 0.04}$ | $\mathbf{0.73}_{\pm 0.10}$ | $\mathbf{0.59}_{\pm 0.05}$ | $\mathbf{0.79}_{\pm 0.04}$ |

ing performance for a single under-represented class out of four. These findings further underscore the importance of generating videos with coherent spatio-temporal context for downstream tool detection models to perform effectively. Similar results were obtained on the GraSP dataset (see Tab.9). *Together, these results highlight that SurgiFlowVid not only improves rare-class detection but also strengthens overall tool recognition performance.*

**Laparoscope motion prediction** Fig. 4 presents the results of laparoscope motion detection on the AutoLaparo dataset. Among the baselines, SurV-Gen (RS) achieves better performance than Endora, while SparseCtrl with RGB-frame conditioning performs best on the online recognition task. Our approach, SurgiFlowVid, already outperforms SurV-Gen with text-only conditioning, and the RGB-mask conditioned version surpasses all baselines. Similar trends

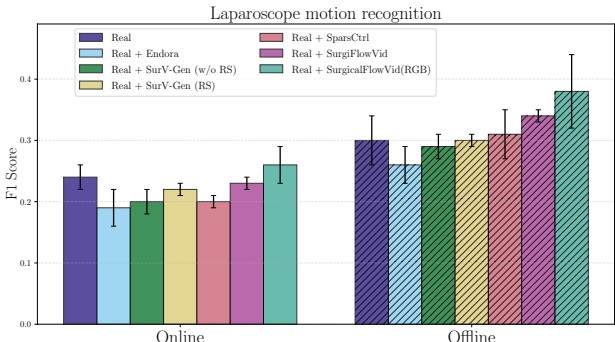

*Figure 4.* **Laparoscope motion prediction** in *online* (left) and *offline* (right) fashion on the AutoLaparo dataset.

*Table 5.* Complexity analysis between models

| Method | Trainable params. (M) | Video resolution | GPU mem (GB) | Inf. time(sec) |
|---|---|---|---|---|
| Endora | 675 | $128 \times 128$ | 22 | 7.85s |
| SurV-Gen | 435 | $256 \times 256$ | 48 | 6.55s |
| SurgiFlowVid | 437 | $512 \times 512$ | 50 | 7.53s |
| SparseCtrl | 453 | $256 \times 256$ | 50 | 10.20s |
| SurgiFlowVid + SVE | 456 | $512 \times 512$ | 52 | 10.45s |

are observed for the offline recognition task, where both F1 scores are higher compared to the online setting. This suggests that providing a longer temporal context enables the downstream model to classify laparoscope motion more accurately. Overall, these findings demonstrate that Surgi-FlowVid can effectively adapt to smaller datasets while offering substantial benefits for developing automatic field-of-view control systems. *This highlights the practical utility of our method in developing real-time surgical assistance systems.*

**Complexity analysis** Compared to Endora, both SurV-Gen and our method utilize fewer trainable parameters (Table 5). Our approach generates videos at $512 \times 512$ resolution, compared to $256 \times 256$ for SurV-Gen and SparseC-trl and $128 \times 128$ for Endora, with only a modest ∼2M-parameter increase from the optical-flow branch. To reduce computational overhead, optical flow maps are pre-computed offline during training and are not used at inference. This overhead is small relative to the consistent performance gains on under-represented classes (up to $+12\%$).

**Sensitivity analysis** We analyze the sensitivity of our method to the optical-flow estimator and the weighting factor $\lambda_p$ by replacing RAFT with UniMatch (Xu et al., 2023) and varying $\lambda_p$ respectively. As reported in Tab. 6, RAFT achieves marginally higher performance, though differences remain small, indicating robustness to the choice of flow model. Performance peaks for $\lambda_p \in [0.5, 0.75]$ ; we therefore fix $\lambda_p = 0.65$ for all experiments based on preliminary validation on a subset dataset.

*Table 6.* Sensitivity analysis to flow estimator and $\lambda_p$.

| Method | A1 | A5 | A7 |
|---|---|---|---|
| RAFT | $0.43_{\pm 0.12}$ | $0.11_{\pm 0.03}$ | $0.35_{\pm 0.12}$ |
| UniMatch | $0.39_{\pm 0.14}$ | $0.10_{\pm 0.02}$ | $0.32_{\pm 0.10}$ |

| $\lambda_p$ | A1 | A5 | A7 |
|---|---|---|---|
| 0.25 | $0.32_{\pm 0.10}$ | $0.09_{\pm 0.05}$ | $0.31_{\pm 0.11}$ |
| 0.50 | $0.43_{\pm 0.12}$ | $0.11_{\pm 0.03}$ | $0.35_{\pm 0.12}$ |
| 0.75 | $0.44_{\pm 0.11}$ | $0.13_{\pm 0.02}$ | $0.33_{\pm 0.09}$ |
| 1.00 | $0.40_{\pm 0.09}$ | $0.10_{\pm 0.04}$ | $0.31_{\pm 0.10}$ |

*Figure 5.* Tool types match the sparse seg. frames, but their position shifts, causing a failure case.

**Limitations** While our approach demonstrates strong performance gains for under-represented classes, it has certain limitations. In this work, we focus on short temporal–context surgical tasks and therefore generate clips of about four seconds. For longer-horizon tasks such as surgical phase recognition, our framework can be extended with autoregressive generation strategies similar to FlowVid (Liang et al., 2024), requiring only minimal modifications to the training setup. Additionally, the sparsity of segmentation frames can occasionally result in incorrect tool positioning (see Fig. 5). Although our method shows better instrument overlap (Sec. B.9) additional gains can be achieved by incorporating tool kinematics or test-time correction, which we leave for future work.

## 6. Conclusion

In this work, we addressed the critical challenge of data imbalance in surgical datasets by generating synthetic video samples of under-represented classes with our proposed framework, *SurgiFlowVid*. The framework generates spatially and temporally coherent videos through a dual-prediction diffusion U-Net that jointly models RGB frames and optical flow, while a sparse visual encoder enables controllable generation using only the limited conditional signals typically available in surgical datasets. Extensive experiments across three datasets and downstream tasks, including surgical action recognition, tool presence detection, and laparoscope motion prediction, demonstrate consistent improvements over strong baselines. By bridging advances in machine learning with the needs of surgical data science, this work helps address the scarcity of data on rare events

and moves toward more robust surgical video understanding models.

## Impact Statement

This work aims to advance machine learning methods for surgical video understanding by addressing data imbalance through the generation of synthetic videos. Improving the robustness of such models has the potential to positively impact healthcare applications by supporting the development of more reliable surgical assistance and analysis systems, particularly for rare but clinically important events. All datasets used in this study are publicly available, and the proposed method is intended as a research tool to augment existing training data rather than to replace clinical expertise or decision-making. We do not foresee direct negative societal impacts arising from this work.

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

**Appendix**

## A. Extended Related Work

**Video diffusion models** Diffusion-based video generation methods have recently demonstrated strong efficiency and scalability by operating in continuous latent spaces (Ho et al., 2020; Rombach et al., 2022a). Early work by (Ho et al., 2022b) extended pixel-space diffusion to videos using probabilistic DMs, while (Harvey et al., 2022) proposed generating sparse frames with interpolation, though limited to low-resolution synthetic datasets. Large-scale efforts such as Make-A-Video (Singer et al., 2023) and Imagen Video (Ho et al., 2022a) employ cascaded super-resolution pipelines built on DALLE-2 (Ramesh et al., 2022) and Imagen (Saharia et al., 2022), respectively, but require billions of parameters and massive compute resources. Stable Video Diffusion (Blattmann et al., 2023) has been widely adopted in the natural image/video community, while several closed-source systems—such as MovieGen (Polyak et al., 2024), Pika[1], Gen (Runway)[2], and Veo[3]—achieve high-quality generation conditioned on diverse modalities ranging from text to depth maps.

On the open-source side, AnimateDiff (Guo et al., 2023) and SparseCtrl (Guo et al., 2024) extend image diffusion models to videos, while OpenSora (Zheng et al., 2024) represents a large community-driven effort to replicate Sora[4]. The CogVideo family (Yang et al., 2024; Hong et al., 2022) introduces expert transformer architectures for video synthesis and has been adopted in prior surgical applications (Biagini et al., 2025; Iliash et al., 2024). However, CogVideo is a 5B parameter model requiring vast datasets and heavy compute, making it impractical for limited surgical data where overfitting is a risk. Similar efforts have been made using large scale open-source videos models like Hunyuan (Wu et al., 2025) and Wan (Wan

---

[1] https://pika.art/login
[2] https://runwayml.com/
[3] https://deepmind.google/models/veo/
[4] https://openai.com/sora/

et al., 2025). We inspire our approach from the more recently proposed methods such as FlowVid (Liang et al., 2024) and VideoJam (Chefer et al., 2025). FlowVid proposed a flow warped video-to-video generation framework, wherein optical flow was used to maintain the structure of objects between frames during translation. This framework trained on a corpus of 10M videos. The primary application of this work differs from ours such that we intend to generate new videos with conditional signals in contrast to performing video-to-video translation. Secondly, VideoJam explored video prediction with a DiT-based architecture (Peebles & Xie, 2023), but its 30B parameter model was trained on 100M videos, produces only $256 \times 256$ outputs, and lacks controllability—an essential requirement for surgical applications.

*Surgical video generation model* like SurgSora (Chen et al., 2025) is a controllable framework for generating surgical videos that incorporates optical flow alongside semantic context, depth maps, and segmentation masks. However, its design differs fundamentally from ours. SurgSora relies on multiple conditioning signals produced by off-the-shelf models that are not trained on surgical video data, and uses a large backbone model (SVD (Rombach et al., 2022b)), which makes it difficult to ensure consistent utilization of all conditioning modalities during generation. Our framework explicitly trains the model to predict optical flow jointly with RGB, ensuring that temporal dynamics are learned directly rather than being implicitly injected. Related efforts in the surgical (He et al., 2025; Shah et al., 2026) or medical world models (Yang et al., 2025a) focus on learning latent actions or simulating disease progression, often using multi-stage pipelines or modalities such as CT, which differ substantially from laparoscopic surgical video streams. Due to differences in conditioning modalities, model capacity, datasets, and task definitions, a direct side-by-side quantitative comparison with such controllable video generation frameworks would require retraining these models from scratch and would not yield a strictly fair evaluation.

In contrast, our work focuses on the surgical domain under constrained compute budgets (1-2 GPUs) and explicitly addresses the challenge of data imbalance. While dual-branch architectures have been explored for surgical understanding tasks, we build upon small-scale surgical video diffusion models and introduce a sparse, controllable generative framework tailored to synthesizing under-represented surgical classes. Although future work may explore scaling to larger models, our approach demonstrates a practical and computationally feasible direction for improving surgical video understanding in realistic healthcare settings.

**Data imbalance**  The presence of rare classes is a common challenge in real-world datasets. In classification, oversampling is frequently used to mitigate this issue by sampling under-represented classes more often during training (Belhaouari et al., 2024). Standard augmentation methods such as horizontal flipping, random resizing, and cropping are widely used, while regional dropout methods (Zhong et al., 2020) randomly remove image regions to improve robustness and generalization. More advanced strategies, including RandAugment (Cubuk et al., 2020) and AutoAugment (Cubuk et al., 2019), apply diverse pixel-level operations (e.g., rotation, shear, translation, color jitter) through either random selection or learned policies. Other approaches combine multiple images, such as Mixup (Zhang et al., 2018a), which blends both pixel values and labels, and CutMix (Yun et al., 2019), which replaces patches from one image with regions from another, maximizing pixel efficiency while mixing labels. These augmentation strategies have been specifically proposed for image classification tasks. Readers can refer to (Chen et al., 2024) for a detailed survey.

Within the surgical domain, class imbalance is particularly prevalent due to challenges in data collection (e.g., reliance on single-center data), the rarity of specific surgical events, and ethical or legal restrictions on data sharing (Salmi et al., 2024; Maier-Hein et al., 2017). Such imbalance often degrades the performance of downstream models. While augmentation and re-sampling strategies have been shown to improve medical imaging tasks (Salmi et al., 2024), surgical video understanding tasks lacks dedicated augmentation approaches. Prior attempts have used synthetic data, for instance via image-to-image translation, to complement real datasets for only surgical instrument segmentation tasks (Colleoni et al., 2022; Colleoni & Stoyanov, 2021; Zhao et al., 2025). In this work, we establish a strong baseline for real datasets by combining curated image-level augmentation techniques with inverse frequency balancing, which up-weights under-represented classes. We use this strategy only during the training of real datasets. To directly assess the utility of synthetic data as a complementary augmentation strategy, we merge generated videos with real data without applying further augmentations.

## B. Additional results

### B.1. Synthetic data attributes

The results on different aspects of synthetic data for the SAR-RARP50 dataset are presented in Tab. 7. Performance remains unchanged when the training data is merely duplicated, a trend consistent across most classes. In contrast, perturbations to either the spatial or temporal structure of the videos result in clear performance degradation. This behavior aligns with

the role of the downstream model, which relies on both spatial structure (e.g., the arrangement of organs) and temporal dynamics (e.g., tissue motion and single or multi-tool interactions) to classify an action. Notably, the action class "cutting the suture," which is already highly imbalanced, suffers a substantial drop in performance when frame-level noise is introduced. Similar results were noticed for the GraSP dataset (Tab. 8). Interestingly we noticed for shuffling the frames lead to a small improvement in scores for two of the under-represented classes. This results could also be attributed to the downstream model architecture difference between the TAPIS model and the plain MViT model. However, overall these findings highlight that synthetic data cannot simply replicate training samples, nor can it exhibit spatial or temporal inconsistencies, if it is to provide meaningful benefits for downstream tasks.

*Table 7.* **Attributes of synthetic** data experiment on the SAR-RARP50 dataset. Merely replicating the training data does not lead to any improvement in performance. The degradation of the spatial or temporal structure leads to a decline in downstream model performance.

| Training data | Pick the needle | Position the needle | Push the needle | Pull the needle | Cut the suture | Return the needle | Mean. |
|---|---|---|---|---|---|---|---|
| Real | $0.32_{\pm 0.19}$ | $0.66_{\pm 0.09}$ | $0.78_{\pm 0.10}$ | $0.61_{\pm 0.09}$ | $0.10_{\pm 0.04}$ | $0.32_{\pm 0.15}$ | $0.46_{\pm 0.08}$ |
| Data duplication | $0.32_{\pm 0.17}$ | $0.60_{\pm 0.03}$ | $0.78_{\pm 0.08}$ | $0.61_{\pm 0.10}$ | $0.10_{\pm 0.03}$ | $0.31_{\pm 0.11}$ | $0.45_{\pm 0.06}$ |
| Frame shuffle | $0.30_{\pm 0.19}$ | $0.63_{\pm 0.08}$ | $0.74_{\pm 0.11}$ | $0.60_{\pm 0.08}$ | $0.06_{\pm 0.09}$ | $0.30_{\pm 0.17}$ | $0.43_{\pm 0.04}$ |
| Sparse frame | $0.28_{\pm 0.14}$ | $0.60_{\pm 0.07}$ | $0.70_{\pm 0.04}$ | $0.59_{\pm 0.09}$ | $0.05_{\pm 0.05}$ | $0.29_{\pm 0.10}$ | $0.42_{\pm 0.03}$ |
| Noisy frame | $0.29_{\pm 0.14}$ | $0.62_{\pm 0.07}$ | $0.76_{\pm 0.04}$ | $0.60_{\pm 0.09}$ | $0.04_{\pm 0.05}$ | $0.29_{\pm 0.10}$ | $0.43_{\pm 0.02}$ |

*Table 8.* **Attributes of synthetic** data experiment on the GraSP dataset.

| Training data | Pull the suture | Tie the suture | Cut the suture | Cut btw.the prostate | Identify iliac artery | Mean. |
|---|---|---|---|---|---|---|
| Real | $0.26_{\pm 0.03}$ | $0.44_{\pm 0.01}$ | $0.43_{\pm 0.06}$ | $0.72_{\pm 0.07}$ | $0.52_{\pm 0.08}$ | $0.46_{\pm 0.08}$ |
| Data duplication | $0.25_{\pm 0.02}$ | $0.44_{\pm 0.02}$ | $0.43_{\pm 0.05}$ | $0.71_{\pm 0.06}$ | $0.52_{\pm 0.04}$ | $0.46_{\pm 0.04}$ |
| Frame shuffle | $0.27_{\pm 0.04}$ | $0.40_{\pm 0.02}$ | $0.42_{\pm 0.01}$ | $0.69_{\pm 0.03}$ | $0.53_{\pm 0.04}$ | $0.46_{\pm 0.02}$ |
| Sparse frame | $0.24_{\pm 0.02}$ | $0.38_{\pm 0.03}$ | $0.40_{\pm 0.02}$ | $0.68_{\pm 0.02}$ | $0.48_{\pm 0.01}$ | $0.43_{\pm 0.02}$ |
| Noisy frame | $0.20_{\pm 0.04}$ | $0.35_{\pm 0.05}$ | $0.34_{\pm 0.06}$ | $0.66_{\pm 0.03}$ | $0.46_{\pm 0.05}$ | $0.40_{\pm 0.04}$ |

### B.2. Additional Surgical action dataset

We further evaluated surgical action recognition on the GynSurg dataset (Nasirihaghighi et al., 2025), which consists of laparoscopic gynecological procedures with four annotated actions: coagulation (P1), needle passing (P2), suction/irrigation (P3), and transection (P4). The classes P3 and P4 are under-represented. Each action is provided as short 3-second video clips, making the dataset well-suited for action recognition. Importantly, this dataset differs substantially from SAR-RARP50 and GraSP in terms of anatomy, environment, tool usage, and camera motion, allowing us to demonstrate the generalizability of our approach across diverse surgical settings. We adopt the MViTv2 model as the downstream architecture.

Results are reported in Fig. 6. Synthetic samples from SparseCtrl improve performance by $8$–$9\%$ for the under-represented classes. In contrast, our method with text conditioning achieves consistent gains across all four classes, raising the average Jaccard score to $0.72$ compared to $0.66$ with real data only. Conditioning with RGB frames yields further improvements of nearly 20 points for P3 and P4. These results highlight the advantage of combining dual-prediction with sparse visual encoding to generate synthetic videos that preserve both spatial and temporal consistency.

### B.3. Surgical tool presence detection

The results on surgical tool presence detection on the GraSP dataset is shown in Tab. 9. The overall addition of synthetic data, leads to improvement in scores for most of the tool classes. The synthetic data from SurgiFlowVid proves useful espscially for the under-represented classes like prograsp forceps and clip applier among others.

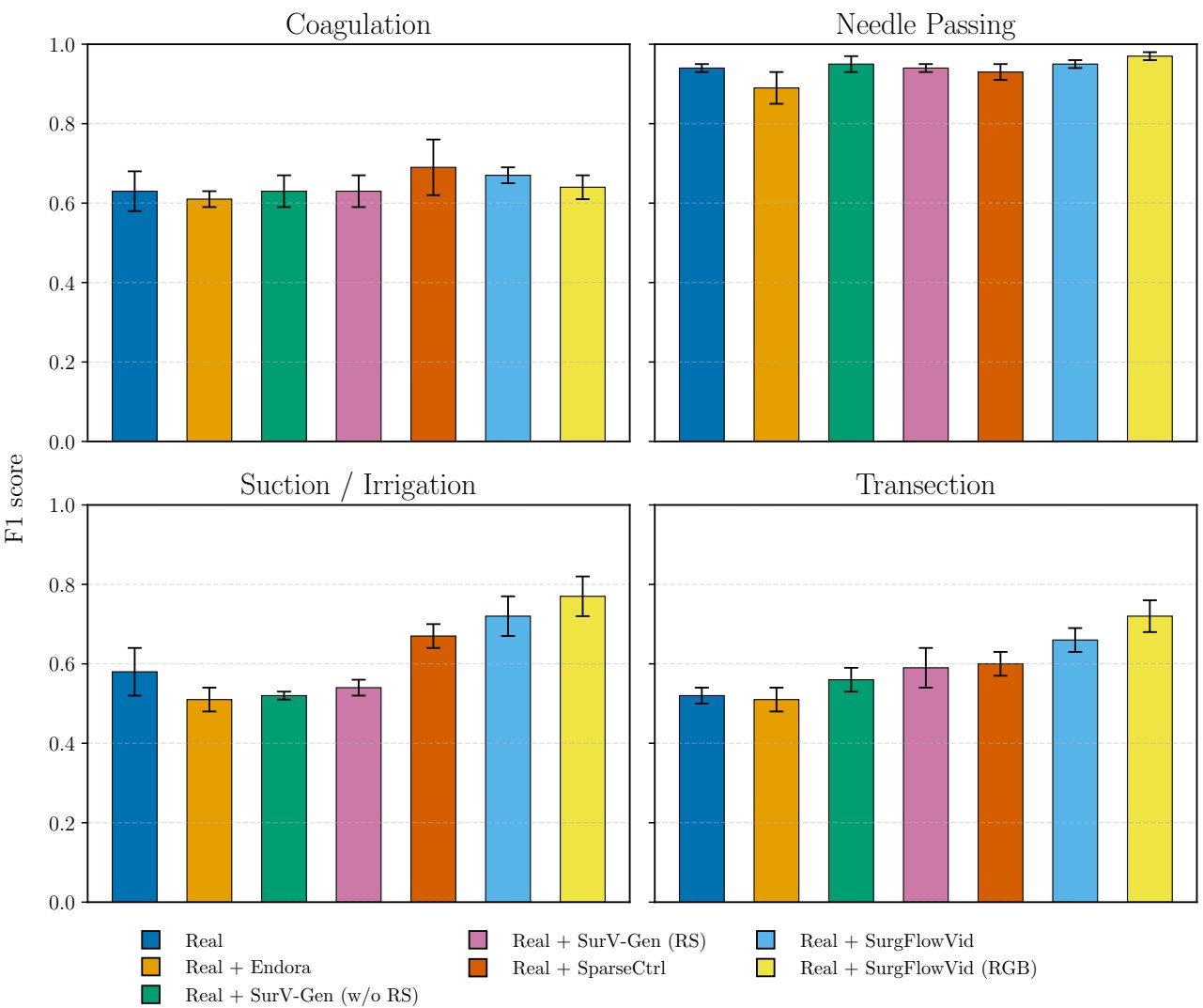

*Figure 6.* **Surgical action recognition** results on the GynSurg dataset, reported using the F1 score. The under-represented classes are "Suction" and "Transection". The addition of synthetic samples for both the balanced classes shows smaller improvements. However, the synthetic video samples from our approach (SurgiFlowVid) with text conditioning improves performance for both under-represented classes, while sparse RGB frame conditioning yields gains of up to 20 points in comparison to using only the real dataset.

*Table 9.* **Surgical tool presence detection on GraSP dataset**. Combining synthetic data from SurgiFlowVid yields marked improvements in dice scores.

| Training data | Bipolar forceps | L.needle driver | Mono curved scissors | Prograsp forceps | Suction inst. | Clip applier | Laparoscopic inst. | Mean |
|---|---|---|---|---|---|---|---|---|
| Real | $0.94_{\pm 0.01}$ | $0.56_{\pm 0.03}$ | $0.95_{\pm 0.02}$ | $0.72_{\pm 0.02}$ | $0.71_{\pm 0.03}$ | $0.34_{\pm 0.09}$ | $0.56_{\pm 0.04}$ | $0.68_{\pm 0.10}$ |
| Real + SparseCtrl(Seg) | $\mathbf{0.95}_{\pm \mathbf{0.02}}$ | $0.56_{\pm 0.02}$ | $0.97_{\pm 0.01}$ | $0.75_{\pm 0.03}$ | $\mathbf{0.74}_{\pm \mathbf{0.07}}$ | $0.35_{\pm 0.02}$ | $\mathbf{0.60}_{\pm \mathbf{0.05}}$ | $0.70_{\pm 0.04}$ |
| Real + SurgFlowVid(Seg) | $0.94_{\pm 0.01}$ | $\mathbf{0.58}_{\pm \mathbf{0.02}}$ | $\mathbf{0.98}_{\pm \mathbf{0.01}}$ | $\mathbf{0.78}_{\pm \mathbf{0.01}}$ | $0.73_{\pm 0.04}$ | $\mathbf{0.37}_{\pm \mathbf{0.03}}$ | $\mathbf{0.60}_{\pm \mathbf{0.02}}$ | $\mathbf{0.72}_{\pm \mathbf{0.02}}$ |

## B.4. Model architecture

We further analyzed the impact of synthetic data using a different architecture for action recognition on SAR-RARP50. We utilized the MViT-v2 (Li et al., 2022) model which is purely transformer-based, we tested whether synthetic samples introduce any architectural bias by comparing against X3D (Feichtenhofer, 2020), a lightweight 3D convolutional model with only 3M parameters (vs. 30M for MViT). The evaluation setup remained identical to previous experiments. The results are shown in Tab. 10. Compared to Tab. 2, the mean Jaccard score with real data dropped to 0.38 for X3D (vs. 0.46 for MViT), as expected given the smaller capacity of X3D.

*Table 10.* **Influence of model architecture**. The surgical action recognition task on the SAR-RARP50 dataset using X3D model. The Jaccard index is reported. Best and second-best scores are highlighted in blue and green, respectively. Under-represented classes are indicated with shade. We notice similar trends to Tab. 2, where the addition of samples from our approach leads to performance gains for all the under-represented classes.

| Training data | Cond. type | | Pick the needle | Position the needle | Push the needle | Pull the needle | Cut the suture | Return the needle | Mean. |
|---|---|---|---|---|---|---|---|---|---|
| | Text | Sparse mask | | | | | | | |
| Real | – | – | $0.22_{\pm0.01}$ | $0.54_{\pm0.08}$ | $0.75_{\pm0.07}$ | $0.51_{\pm0.13}$ | $0.10_{\pm0.02}$ | $0.20_{\pm0.12}$ | $0.38_{\pm0.06}$ |
| Real + Endora | – | – | $0.19_{\pm0.04}$ | $0.53_{\pm0.02}$ | $0.75_{\pm0.05}$ | $0.50_{\pm0.10}$ | $0.09_{\pm0.05}$ | $0.18_{\pm0.04}$ | $0.38_{\pm0.06}$ |
| Real + SurV-Gen (w/o RS) | – | – | $0.22_{\pm0.10}$ | $0.54_{\pm0.04}$ | $0.75_{\pm0.02}$ | $0.51_{\pm0.08}$ | $0.11_{\pm0.09}$ | $0.19_{\pm0.08}$ | $0.39_{\pm0.07}$ |
| Real + SurV-Gen (RS) | – | – | $0.23_{\pm0.11}$ | $0.54_{\pm0.06}$ | $0.74_{\pm0.07}$ | $0.52_{\pm0.11}$ | $0.10_{\pm0.09}$ | $0.23_{\pm0.16}$ | $0.39_{\pm0.06}$ |
| Real + SparseCtrl | | RGB | $0.34_{\pm0.17}$ | $0.60_{\pm0.07}$ | $0.77_{\pm0.08}$ | $0.58_{\pm0.09}$ | $0.08_{\pm0.05}$ | $0.23_{\pm0.16}$ | $0.43_{\pm0.03}$ |
| Real + SparseCtrl | | Seg. | $0.33_{\pm0.14}$ | $0.58_{\pm0.06}$ | $0.75_{\pm0.07}$ | $0.57_{\pm0.13}$ | $0.09_{\pm0.03}$ | $0.28_{\pm0.17}$ | $0.43_{\pm0.04}$ |
| Real + SurgFlowVid | | – | $0.34_{\pm0.13}$ | $0.58_{\pm0.06}$ | $0.75_{\pm0.05}$ | $0.55_{\pm0.13}$ | $0.18_{\pm0.09}$ | $0.29_{\pm0.12}$ | $0.45_{\pm0.04}$ |
| Real + SurgFlowvid | | RGB | $0.30_{\pm0.19}$ | $0.58_{\pm0.06}$ | $0.74_{\pm0.07}$ | $0.58_{\pm0.10}$ | $0.10_{\pm0.08}$ | $0.26_{\pm0.17}$ | $0.43_{\pm0.02}$ |
| Real + SurgFlowVid | | Seg. | $0.39_{\pm0.12}$ | $0.60_{\pm0.05}$ | $0.76_{\pm0.08}$ | $0.56_{\pm0.12}$ | $0.13_{\pm0.05}$ | $0.35_{\pm0.12}$ | $0.47_{\pm0.02}$ |

Synthetic data from SparseCtrl led to modest improvements, while SurgiFlowVid with text conditioning provided only subtle gains. However, consistent with trends in Tab. 2, adding sparse RGB or segmentation masks as conditional signals in SurgiFlowVid yielded considerable improvements across the under-represented classes. Similar trends were noticed when we performed individual class modelling with the results shown in Tab. 11. These findings suggest that performance gains from synthetic data are not biased toward a specific architecture; instead, both transformer- and convolution-based models benefit from the spatial and temporal consistency encoded in synthetic videos. For the GraSP dataset, we opted to use the TAPIS model as proposed in (Ayobi et al., 2025) as this model performed in par with other convolutional architectues.

Using the features extracted from downstream models, temporal models are trained to enhance action recognition further. However, the reported performance improvements were minimal (Funke et al., 2025), and we therefore did not pursue such experiments in this study. Future work could explore this direction in greater depth, focusing on identifying which features from synthetic data are most beneficial for improving the generation process. Additionally, incorporating temporal learning strategies on top of these features may provide further gains for surgical action recognition tasks.

*Table 11.* **Influence of model architecture**. The surgical action recognition task on the SAR-RARP50 dataset using X3D model with *individual class modelling*. The Jaccard index is reported. We notice smaller gains for the action "cut the suture" (see Tab. 10) by modeling each of the under-represented classes separately.

| Training data | Cond. type | | Pick the needle | Position the needle | Push the needle | Pull the needle | Cut the suture | Return the needle | Mean. |
|---|---|---|---|---|---|---|---|---|---|
| | Text | Sparse mask | | | | | | | |
| Real | – | – | $0.22_{\pm0.01}$ | $0.54_{\pm0.08}$ | $0.75_{\pm0.07}$ | $0.51_{\pm0.13}$ | $0.10_{\pm0.02}$ | $0.20_{\pm0.12}$ | $0.38_{\pm0.06}$ |
| Real + SurV-Gen (RS) | | – | $0.25_{\pm0.12}$ | $0.54_{\pm0.03}$ | $0.76_{\pm0.09}$ | $0.51_{\pm0.09}$ | $0.10_{\pm0.13}$ | $0.24_{\pm0.18}$ | $0.40_{\pm0.05}$ |
| Real + SparseCtrl | | RGB | $0.30_{\pm0.16}$ | $0.59_{\pm0.07}$ | $0.75_{\pm0.06}$ | $0.57_{\pm0.11}$ | $0.10_{\pm0.09}$ | $0.21_{\pm0.12}$ | $0.42_{\pm0.03}$ |
| Real + SparseCtrl | | Seg. | $0.30_{\pm0.17}$ | $0.57_{\pm0.04}$ | $0.76_{\pm0.07}$ | $0.57_{\pm0.09}$ | $0.20_{\pm0.05}$ | $0.37_{\pm0.10}$ | $0.46_{\pm0.01}$ |
| Real + SurgFlowvid | | RGB | $0.40_{\pm0.16}$ | $0.56_{\pm0.02}$ | $0.75_{\pm0.04}$ | $0.56_{\pm0.16}$ | $0.23_{\pm0.13}$ | $0.35_{\pm0.15}$ | $0.48_{\pm0.02}$ |
| Real + SurgFlowVid | | Seg. | $0.39_{\pm0.11}$ | $0.59_{\pm0.04}$ | $0.77_{\pm0.03}$ | $0.55_{\pm0.10}$ | $0.15_{\pm0.06}$ | $0.40_{\pm0.10}$ | $0.48_{\pm0.05}$ |

## B.5. Video metrics

We assess the temporal performance of the model using *Segmental F1@K* score. This metric penalizes both out-of-order predictions and over-segmentation. Segmental F1@K quantifies the temporal overlap between predicted and ground-truth segments, while being less sensitive to small boundary shifts caused by annotation noise. The metric is defined as,

$$\text{SegmentalF1@K} = \frac{2 \times (\text{Pr} \times \text{Rc})}{(\text{Pr} + \text{Rc})}, \tag{3}$$

where Pr and Rc denotes precision and recall. A prediction is considered a true positive (TP) if the IoU exceeds the threshold $T = K/100$; otherwise, it is counted as a false positive (FP). The results of the recognition task are shown in Tab.12 and Tab.13. Compared to using only the real dataset, the addition of synthetic samples leads to smaller improvements in overall performance. The addition of either RGB or segmentation conditioning lead to a similar scores of 0.37 and 0.36 respectively. Overall, the synthetic samples from SurgiFlowVid prove very beneficial for both the balanced and the under-represented classes.

*Table 12.* **Surgical action recognition** on the SAR-RARP50 dataset. Segmental F1 scores are reported.

| Training data | Cond. type | | Pick the needle | Position the needle | Push the needle | Pull the needle | Cut the suture | Return the needle | Mean. |
|---|---|---|---|---|---|---|---|---|---|
| | Text | Sparse mask | | | | | | | |
| Real | – | – | $0.28_{\pm 0.17}$ | $0.40_{\pm 0.16}$ | $0.62_{\pm 0.18}$ | $0.41_{\pm 0.14}$ | $0.09_{\pm 0.08}$ | $0.22_{\pm 0.18}$ | $0.32_{\pm 0.06}$ |
| Real + Endora | – | – | $0.23_{\pm 0.13}$ | $0.38_{\pm 0.06}$ | $0.55_{\pm 0.09}$ | $0.41_{\pm 0.10}$ | $0.09_{\pm 0.09}$ | $0.21_{\pm 0.08}$ | $0.31_{\pm 0.08}$ |
| Real + SurV-Gen (w/o RS) | – | – | $0.26_{\pm 0.12}$ | $0.40_{\pm 0.04}$ | $0.55_{\pm 0.08}$ | $0.41_{\pm 0.06}$ | $0.12_{\pm 0.09}$ | $0.23_{\pm 0.12}$ | $0.33_{\pm 0.04}$ |
| Real + SurV-Gen (RS) | – | – | $0.27_{\pm 0.14}$ | $0.40_{\pm 0.15}$ | $0.58_{\pm 0.19}$ | $0.42_{\pm 0.18}$ | $0.20_{\pm 0.13}$ | $0.23_{\pm 0.18}$ | $0.35_{\pm 0.07}$ |
| Real + SparseCtrl | | RGB | $0.32_{\pm 0.20}$ | $0.41_{\pm 0.16}$ | $0.57_{\pm 0.17}$ | $0.44_{\pm 0.15}$ | $0.10_{\pm 0.09}$ | $0.25_{\pm 0.11}$ | $0.35_{\pm 0.03}$ |
| Real + SurgFlowVid | | – | $0.27_{\pm 0.14}$ | $0.40_{\pm 0.16}$ | $0.57_{\pm 0.16}$ | $0.43_{\pm 0.13}$ | $0.13_{\pm 0.08}$ | $0.16_{\pm 0.07}$ | $0.33_{\pm 0.04}$ |
| Real + SurgFlowvid | | RGB | $0.31_{\pm 0.17}$ | $0.43_{\pm 0.17}$ | $0.59_{\pm 0.16}$ | $0.45_{\pm 0.10}$ | $0.15_{\pm 0.04}$ | $0.31_{\pm 0.12}$ | $0.37_{\pm 0.03}$ |

*Table 13.* **Surgical action recognition** on the SAR-RARP50 dataset. Segmental F1 scores are reported. for seg. frame conditioning.

| Training data | Cond. type | | Pick the needle | Position the needle | Push the needle | Pull the needle | Cut the suture | Return the needle | Mean. |
|---|---|---|---|---|---|---|---|---|---|
| | Text | Sparse mask | | | | | | | |
| Real | – | – | $0.28_{\pm 0.17}$ | $0.40_{\pm 0.16}$ | $0.62_{\pm 0.18}$ | $0.41_{\pm 0.14}$ | $0.09_{\pm 0.08}$ | $0.22_{\pm 0.18}$ | $0.32_{\pm 0.06}$ |
| Real + SparseCtrl | | Seg | $0.33_{\pm 0.19}$ | $0.43_{\pm 0.14}$ | $0.60_{\pm 0.19}$ | $0.44_{\pm 0.15}$ | $0.12_{\pm 0.10}$ | $0.20_{\pm 0.10}$ | $0.35_{\pm 0.05}$ |
| Real + SurgFlowvid | | Seg | $0.30_{\pm 0.14}$ | $0.42_{\pm 0.16}$ | $0.58_{\pm 0.14}$ | $0.43_{\pm 0.13}$ | $0.13_{\pm 0.08}$ | $0.32_{\pm 0.11}$ | $0.36_{\pm 0.02}$ |

## B.6. Ablation on sparse frames

We conducted an ablation study to examine the effect of the number of sparse RGB frames used during generation. We hypothesized that too few frames would provide insufficient controllability, while too many would replicate training data, reducing diversity. To test this, we varied the number of conditioning frames $(1, 3, 5, 10, 12)$ and generated videos, comparing their performance against models trained solely on real data. Results are shown in Fig. 7 (all minor classes modeled jointly) and Fig. 8 (each class modeled separately). A consistent trend across both settings is that using only one frame yields performance similar to the real-only baseline, indicating limited consistency and, in some cases, degenerate generations. Conversely, conditioning on 12 of the 16 frames produced results close to the real dataset baseline, as little additional diversity was introduced. Based on these findings, we adopted a strategy of sampling 3–5 random frames from the real dataset as conditional inputs. These experiments were initially conducted with the X3D model, and the same frame distribution was subsequently applied across all experiments, including the SparseCtrl baseline.

## B.7. Image Quality Analysis

As our goal is to mitigate data imbalance, we focused primarily on generating videos of under-represented classes and evaluating them on the downstream task. We consider this approach as an effective way to directly measure the effectiveness

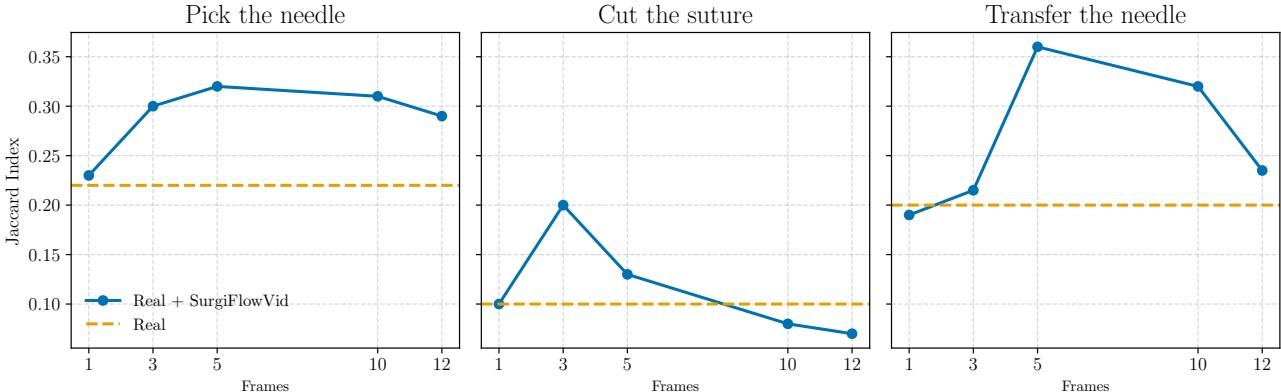

*Figure 7.* **Frame ablation**. The ablation on the number of sparse RGB frames on the SAR-RARP50 dataset. The results consists of using a X3D model with all the minor classes modeled together.

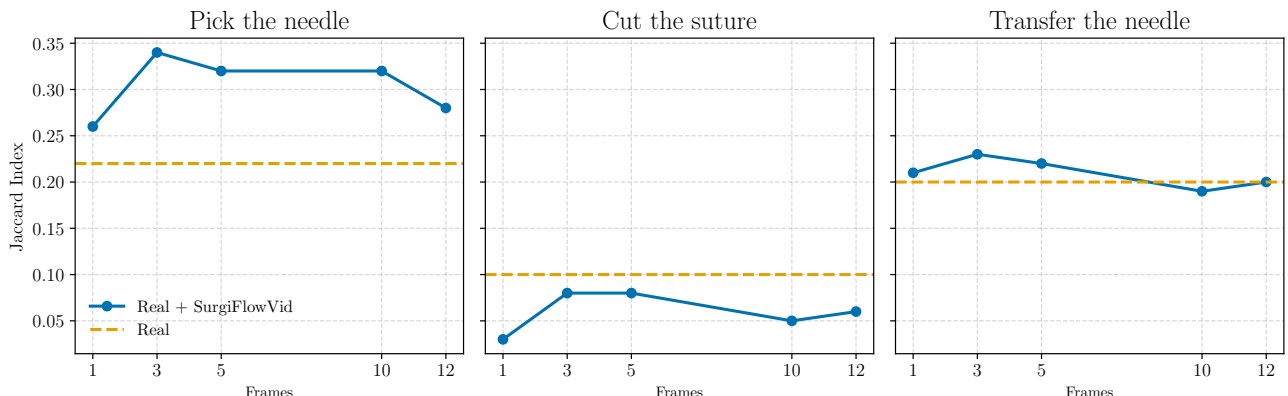

*Figure 8.* **Frame ablation**. The ablation on the number of sparse RGB frames on the SAR-RARP50 dataset. The results consists of using a X3D model with all the minor classes modeled separately.

*Table 14.* **Image quality metrics**. The CLIP image score of different methods are reported here. Higher is better.

| Method | SAR-RARP50 | | | GynSurg | | GraSP | | | |
| --- | --- | --- | --- | --- | --- | --- | --- | --- | --- |
| | A1 | A5 | A7 | P3 | P4 | G1 | G2 | G3 | G4 |
| Endora | 70.30 | 66.85 | 73.65 | 69.43 | 70.12 | 57.09 | 68.10 | 74.41 | 60.72 |
| SurV-Gen | 75.30 | 70.22 | 78.85 | 71.30 | 75.83 | 62.15 | 73.10 | 68.32 | 62.15 |
| SurgiFlowVid | 74.46 | 76.08 | 78.25 | 72.95 | 66.76 | 68.20 | 70.10 | 72.15 | 65.27 |

and the usefulness of the synthetic videos. In this section, we evaluate the quality of the generated videos with the CLIP (Hessel et al., 2021) image and the LPIPS (Zhang et al., 2018b) score. Both these metrics evaluate the quality of the generated frames using features from pre-trained models on large-scale natural images. The results are shown in Tab. 14 and Tab. 15. We compare our approach, SurgiFlowVid with text conditioning against Endora and SurV-Gen. We do not compute these scores for SparseCtrl or sparse visual encoder using our approach, as there already exists frames from the real dataset. The image quality varied between different classes and we did not notice a co-relation between these scores to the downstream model performance. Hence, these values should be interpreted with caution given that they are computed with pre-trained weights from models not trained on surgical images/videos.

*Table 15.* **Image quality metrics**. The LPIPS score of different methods are reported here. Lower is better.

| Method | SAR-RARP50 | | | GynSurg | | GraSP | | | |
| --- | --- | --- | --- | --- | --- | --- | --- | --- | --- |
| | A1 | A5 | A7 | P3 | P4 | G1 | G2 | G3 | G4 |
| Endora | 0.70 | 0.53 | 0.59 | 0.54 | 0.49 | 0.63 | 0.66 | 0.65 | 0.63 |
| SurV-Gen | 0.68 | 0.54 | 0.57 | 0.51 | 0.56 | 0.57 | 0.67 | 0.71 | 0.74 |
| SurgiFlowVid | 0.66 | 0.56 | 0.52 | 0.49 | 0.50 | 0.51 | 0.60 | 0.74 | 0.72 |

## B.8. Motion diversity

We quantify the intra-set video diversity using the VJEPA embeddings for the generated videos. Particularly, we measure the diversity within the generated videos using the video-level embeddings from the pre-trained VJEPA (Assran et al., 2025) model. The mean pairwise cosine distance (scores) is computed between the embeddings and the scores of the real dataset serve as the upper bound. A higher value indicates larger diversity within the generated videos.

*Table 16.* **Motion diversity**. The cosine distance between the video embedding from the VJEPA2 model is reported. Higher value indicates better diversity.

| Method | SAR-RARP50 | | | GraSP | | | |
| --- | --- | --- | --- | --- | --- | --- | --- |
| | A1 | A5 | A7 | G1 | G2 | G3 | G4 |
| Real | 0.121 | 0.115 | 0.109 | 0.102 | 0.113 | 0.131 | 0.117 |
| Endora | 0.043 | 0.061 | 0.045 | 0.064 | 0.079 | 0.056 | 0.065 |
| SurV-Gen | 0.056 | 0.102 | 0.048 | 0.056 | 0.094 | 0.067 | 0.072 |
| SurgiFlowVid | 0.113 | 0.110 | 0.101 | 0.091 | 0.083 | 0.106 | 0.110 |

As shown in the Tab. 16, SurgiFlowVid consistently produces higher scores which supports our claim that the proposed method generates diverse videos for challenging, under-represented classes while maintaining realism.

## B.9. Mask overlap of generated videos

In Fig. 5, we observe that videos generated with sparse segmentation frames sometimes exhibit a drift in tool position relative to the provided masks. We attribute this behaviour to the high sparsity typical of surgical datasets.

To further quantify this effect, we compare the mask overlap between the generated frames and the ground-truth masks. For this analysis, we train a SegFormer (Xie et al., 2021) model in a binary segmentation setting and evaluate videos produced by both SurgiFlowVid and SparseCtrl. As shown in Tab. 17, IoU scores are low for both models, reflecting the difficulty of the task, but our method leads by +8 points over SparseCtrl. This indicates that the dual-prediction design of SurgiFlowVid better preserves tool position.

*Table 17.* **Mask overlap of generated videos**.

| Method | IOU ↑ |
| --- | --- |
| SparseCtrl | 0.34 |
| SurgiFlowVid | 0.42 |

Nonetheless, there remains room for improvement, for example, by incorporating the strategies discussed in the limitations

section.

## B.10. Video augmentation baselines

In this work, we use synthetic videos as a form of data augmentation to reduce class imbalance in the real dataset. To compare against strong augmentation strategies, we conducted additional experiments using VideoMix-Spatial (Yun et al., 2020) and TubeMix (Yun et al., 2020) for action recognition on the SAR-RARP50 dataset. The results in Tab. 18 indicate that neither approach yielded substantial improvements, underscoring the need for more targeted augmentation methods such as the generative strategy proposed here. Overall, SurgiFlowVid achieves the highest performance on the under-represented classes.

*Table 18.* **video augmentation baselines** experiment on the SAR-RARP50 dataset. The addition of synthetic data proves the most useful in comparison to other data augmentations.

| Training data | Pick the needle | Position the needle | Push the needle | Pull the needle | Cut the suture | Return the needle | Mean. |
|---|---|---|---|---|---|---|---|
| Real + VideoMix | $0.33_{\pm 0.14}$ | $0.64_{\pm 0.02}$ | $0.77_{\pm 0.02}$ | $0.61_{\pm 0.04}$ | $0.10_{\pm 0.03}$ | $0.30_{\pm 0.13}$ | $0.45_{\pm 0.03}$ |
| Real + TubeMix | $0.31_{\pm 0.17}$ | $0.65_{\pm 0.03}$ | $0.78_{\pm 0.05}$ | $0.59_{\pm 0.02}$ | $0.09_{\pm 0.05}$ | $0.27_{\pm 0.16}$ | $0.45_{\pm 0.04}$ |
| Real + SurgiFlowVid | $0.44_{\pm 0.18}$ | $0.66_{\pm 0.07}$ | $0.79_{\pm 0.08}$ | $0.64_{\pm 0.04}$ | $0.18_{\pm 0.09}$ | $0.42_{\pm 0.12}$ | $0.52_{\pm 0.04}$ |

## B.11. Laparoscope motion

In addition to the F1 score, we also computed the balanced accuracy as an additional metric. Fig. 9 shows the results on the laparoscope motion prediction task. Similar to the results seen in Fig. 4, the overall scores are higher for the the offline recognition.

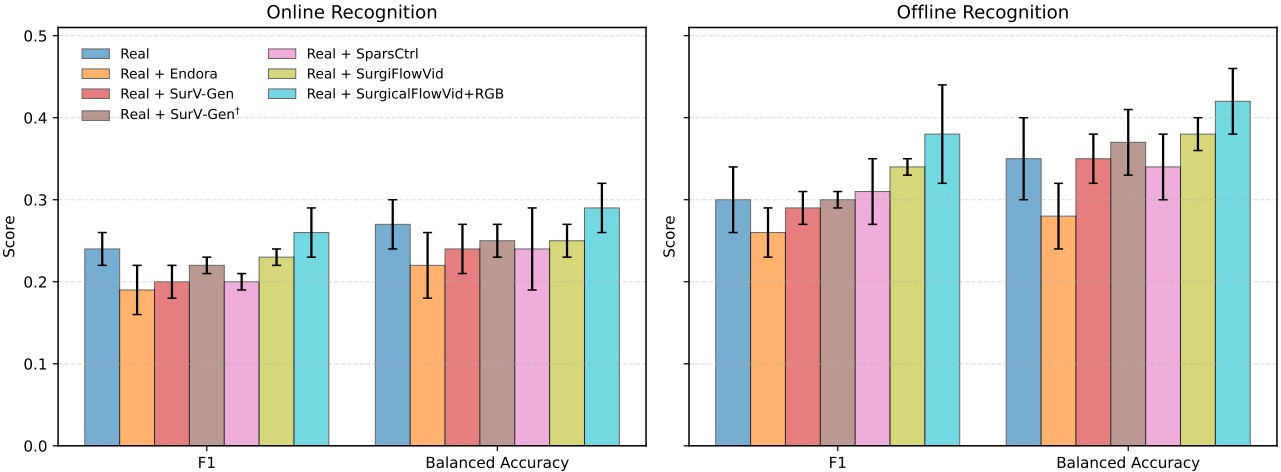

*Figure 9.* Laparoscope motion prediction on the Autolaparo dataset. Bars show mean score with standard deviation (error bars).

## B.12. Model Analysis

In this section, we analyze the model in terms of the video generation cost. The results are shown in Tab. 19. In comparison to Endora, both SurgV-Gen and our approach have lesser number of training parameters as the training is conducted in different stages. Our method, SurgiFlowVid is capable of generating videos at the resolution of $512 \times 512$ pixels whereas the baselines, SurV-Gen and SparseCtrl generates videos at $256 \times 256$ pixels and Endora at $128 \times 128$ pixels. We also train our approach at $512 \times 512$ pixels. Our framework is capable of training at lower resolutions but we opted to train them at higher resolutions as it could be helpful for the downstream task. There exists certain organs or auxiliary tool structures which appears to be very small in shape. Generating videos at higher resolution can benefit these downstream models to learn these

spatial structures effectively. We noticed the benefits for the classification of *catheter* and *clamps* in SAR-RARP50 dataset with synthetic videos from SurgiFlowVid (see Tab. 4). However, an analysis on the video resolution for the downstream task could shed more insights and we leave that for future work. As we generate videos at higher resolution, our approach requires a small overhead in terms of training and sampling times. We believe with the innovations in high performant GPUs these costs could be lowered drastically.

*Table 19.* **Model analysis**. The various parameters of the different baselines. SVE denotes the sparse visual encoder in our approach. The inference time was measured on a A100-80GB GPU.

| Method | Trainable params. (M) | Video resolution | Sampling steps | Inf. time(sec) |
|---|---|---|---|---|
| Endora | 675 | $128 \times 128$ | 50 | 7.85s |
| SurV-Gen | 435 | $256 \times 256$ | 50 | 6.55s |
| SurgiFlowVid | 437 | $512 \times 512$ | 50 | 7.53s |
| SparseCtrl | 453 | $256 \times 256$ | 30 | 10.20s |
| SurgiFlowVid + SVE | 456 | $512 \times 512$ | 30 | 10.45s |

### B.13. Influence of Pre-Training Data

A large scale in-house dataset was used as the pretraining data. We hypothesize that such large scale pre-training can reduce the overall training time on the downstream data as the necessary temporal motions within the surgical videos can be learnt during pre-training.

To decouple the dual-prediction module's influence from the pre-training data, we conducted an additional ablation study on the SAR-RARP50 dataset with no pre-training. We did not notice a large difference in performance, while the number of training steps needed increased $2\times$ for training from scratch.

*Table 20.* **Ablation on pre-training**. Performance on SAR-RARP50 with and without pre-training.

| Train data | Pre-training | Num. train steps | A1 | A2 | A3 | A4 | A5 | A6 | Mean |
|---|---|---|---|---|---|---|---|---|---|
| SurgiFlowVid | ✓ | 90K | $0.43_{\pm 0.12}$ | $0.65_{\pm 0.07}$ | $0.77_{\pm 0.07}$ | $0.63_{\pm 0.11}$ | $0.11_{\pm 0.03}$ | $0.35_{\pm 0.12}$ | $0.49_{\pm 0.04}$ |
| SurgiFlowVid | ✗ | 170K | $0.42_{\pm 0.10}$ | $0.66_{\pm 0.06}$ | $0.77_{\pm 0.03}$ | $0.64_{\pm 0.12}$ | $0.10_{\pm 0.02}$ | $0.36_{\pm 0.11}$ | $0.49_{\pm 0.06}$ |

As shown Tab. 20, training the model from scratch required 170k steps to converge, nearly **double the training steps** to achieve a comparable score to the pre-trained model. These results show that the dual-prediction module is the primary source of improvement. We ensure fair comparison by using these pre-trained checkpoints as a consistent starting point for all baselines. While pre-training accelerates the learning of spatio-temporal features (Rombach et al., 2022b), it is not a prerequisite for achieving the reported gains.

### B.14. Resolution-Controlled Ablation Study

The reported baselines each generates videos at various resolutions. We conduct an ablation study to analyze the influence of the video resolution on its impact on the downstream task. At matched resolution, SurgiFlowVid (text-only) shows comparable performance to SurV-Gen with a notable gain for the "pick the needle" class, suggesting the *optical flow (dual-prediction module) provides meaningful inductive biases* for certain under-represented classes.

As a note, for downstream model training, we resize all the videos to the same resolution. The modest overall difference at $256 \times 256$ indicates that both the optical flow and higher resolution jointly contribute to the gains in Tab.21 (text-only). The dual-prediction module independently improves performance, while higher resolution offers complementary benefits by preserving fine-grained surgical details such as needles and thread.

*Table 21.* **Resolution-controlled ablation**. Comparison at matched $256 \times 256$ resolution and at native $512 \times 512$ resolution.

| Training data | Resolution | A1 | A2 | A3 | A4 | A5 | A6 | Mean |
|---|---|---|---|---|---|---|---|---|
| SurV-Gen | $256 \times 256$ | $0.31_{\pm 0.19}$ | $0.64_{\pm 0.07}$ | $0.77_{\pm 0.06}$ | $0.60_{\pm 0.10}$ | $0.13_{\pm 0.10}$ | $0.37_{\pm 0.18}$ | $0.46_{\pm 0.03}$ |
| SurgiFlowVid | $256 \times 256$ | $0.40_{\pm 0.09}$ | $0.63_{\pm 0.04}$ | $0.77_{\pm 0.06}$ | $0.61_{\pm 0.08}$ | $0.10_{\pm 0.04}$ | $0.34_{\pm 0.11}$ | $0.47_{\pm 0.02}$ |
| SurgiFlowVid | $512 \times 512$ | $0.43_{\pm 0.12}$ | $0.65_{\pm 0.07}$ | $0.77_{\pm 0.07}$ | $0.63_{\pm 0.11}$ | $0.11_{\pm 0.03}$ | $0.35_{\pm 0.12}$ | $0.49_{\pm 0.04}$ |

## C. Dataset

SAR-RARP50: The dataset consists actions annotated at 10 fps. Our initial experiments indicated this temporal frame to be very fine and hence we chose to sample the frames at 5 fps. The annotations for the surgical tools were available at 1 fps making it sparse in nature. For the sparse conditional generation, we randomly samples video frames in the range $3 - 5$ and place them in a different temporal order than the real dataset, so as to create the synthetic data as diverse as possible. For the sparse segmentation conditioning, we opted to include a minimum of 4 frames in the 16 frames video clips during training and sampling time.

GraSP: This dataset consists of annotations at both 30 and 1 fps temporal windows. As 1fps was very coarse in nature, we opted to sample frames at 5 fps from the 30 fps annotations. The segmentation annotations were available at every 35 seconds making them very sparse in nature. Based on dataset analysis, we noticed that creating video clips with at least one segmentation frame as conditioning for the under-represented samples were very challenging. Hence, we opted out of segmentation frames conditioning for the sparse visual encoder in our experiments. However, for the surgical tool presence detection task, we sampled a minimum of 4 frames around the available segmentation frame and used it as the conditioning to generate videos for this task.

The details on the addition of synthetic samples are shown in Tab. 22. We compute the imbalance ratio as the number of clips for the well balanced (most represented) class divided by the number of clips for the other classes. For the SAR-RARP50 dataset, we chose the classes with the ratio higher than 2. Similarly, for the GRaSP and GynSurg datset, the ratio was chosen as 1.5 respectively.

*Table 22.* **Dataset details**. The values in the table include the total number of video clips from the training set. We add only synthetic samples to the under-represented classes to match and balance the instances with the well balanced classes.

| Dataset | Step/action class | Data points in real dataset | Added syn. samples | Imbalance ratio |
|---|---|---|---|---|
| SAR-RARP50 | Pick the needle | 332 | 900 | 4.20 |
| | Position the needle | 1329 | - | 1.04 |
| | Push the needle | 1395 | - | 1.00 |
| | Pull the needle | 1208 | - | 1.15 |
| | Cut the suture | 115 | 1100 | 12.13 |
| | Return the needle | 168 | 1100 | 8.30 |
| GraSP | Pull the suture | 992 | 1600 | 2.82 |
| | Tie the suture | 712 | 1800 | 3.93 |
| | Cut the suture | 1213 | 1300 | 2.30 |
| | Cut btw. the prostate | 1616 | 1000 | 1.73 |
| | Identify iliac artery | 2800 | - | 1.00 |
| GynSurg | Coagulation | 690 | - | 1.25 |
| | Needle passing | 869 | - | 1.00 |
| | Suction/Irrigation | 267 | 550 | 3.25 |
| | Transection | 168 | 650 | 5.17 |

# D. Model training

## D.1. Diffusion Image pre-training

We build upon the SurV-Gen model (Venkatesh et al., 2025a), which was initially proposed to generate synthetic samples of under-represented classes to mitigate data imbalance in surgical datasets. The framework adopts a multi-stage training procedure. In the first stage, frames are extracted from the training split of surgical videos and a 2D Stable Diffusion (SD) model (Rombach et al., 2022a) is trained. We follow the same pipeline with several modifications. Training the spatial SD directly on the limited frames from the downstream task datasets can result in overfitting, reduced diversity of generated frames, or potential data leakage. This phenomenon was observed in SurV-Gen, where synthetic augmentation yielded only marginal improvements without rejection sampling.

To address this issue, we curated an in-house dataset comprising video recordings from different surgical procedures. The dataset consists of approximately 7000 clips, each ranging from 6 to 8 minutes in length. From this collection, we extracted $\sim 4000$ frames to train the 2D component of the model. We initialized training from the SD-v1.5 checkpoint, pre-trained on the large-scale LAION-5B dataset (Schuhmann et al., 2022), which provided a strong initialization compared to training from scratch. The model was fine-tuned for 3000 steps using the AdamW optimizer (Loshchilov & Hutter, 2017) with a learning rate of $1e^{-4}$, a batch size of 2, and gradient checkpointing enabled. Due to computational constraints, frames were resized from their original resolution of $1048 \times 2048$ to $512 \times 512$. For text conditioning, we employed simple prompts such as *"An image of a surgical procedure"*, with embeddings generated using the CLIP text encoder (Radford et al., 2021). This fine-tuned SD model served as the base 2D diffusion prior for any subsqent 2D diffusion models. We fine-tune this model on the downstream datasets before video diffusion training. The spatial priors are learnt during this stage.

## D.2. Diffusion Video pre-training

Next, we focus on the video training stage. In the SurV-Gen approach, the spatial layers are frozen and only the temporal attention layers are trained during the second stage. In contrast, our framework trains the temporal layers jointly with both RGB and optical flow frames. To further improve temporal modeling, we investigated a video pre-training strategy inspired by previous works on video diffusion models (Rombach et al., 2022b; Polyak et al., 2024). Our hypothesis is that temporal motion priors, such as the movement of tools, tissue motions andpartially tool tissue interactions can be better learned by training on the unconditional internally curated dataset, which contains diverse anatomical structures, varying illumination conditions, different endoscope motions, and a wide range of surgical tools and tool interactions. This dataset introduces substantial variability that more closely reflects real-world surgical scenarios.

To test this, we extended SurV-Gen and trained it in two ways, keeping the training recipe unchanged (i.e., only the temporal attention layers are updated). First, we trained SurV-Gen directly on the SAR-RARP50 dataset, where the 2D SD backbone was also trained on frames extracted from the same dataset. Second, we replaced the 2D SD backbone with our fine-tuned 2D model and pre-trained the temporal layers on the curated dataset of $\sim 7000$ videos. For this, we created overlapping subsets of 3000, 5000, and 7000 videos, each containing at least 1500 new clips. The pre-trained temporal layers were then fine-tuned on SAR-RARP50.

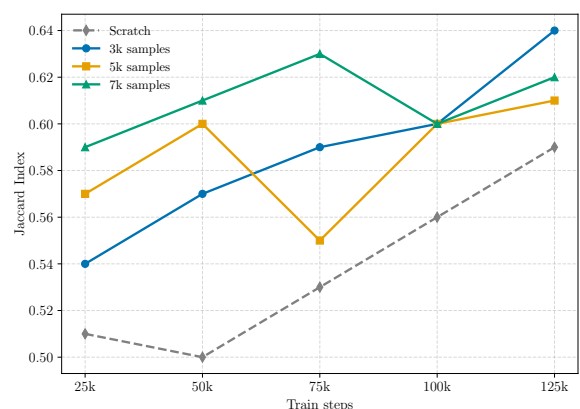

*Figure 10.* The results on video pre-training.

This pre-training strategy is expected to accelerate learning of spatio-temporal representations from the limited SAR-RARP50 data. We then generated synthetic samples of under-represented classes using label guidance, following SurV-Gen, and evaluated their impact on downstream action recognition performance. The results are shown in Fig. 10. We analyzed only the three under-represented classes and report the weighted average Jaccard index of these classes. We notice that the pre-training strategy leads to higher recognition scores in comparison to using only the real dataset for the same number of training steps. We noticed smaller dips in performance for the 5k and 7k samples, which could be attributed to a distributional shift to the SAR-RARP50 dataset. On the other hand, we noticed a continuous improvement in jaccard

scores for the 3k samples. Overall, these results indicate that the pre-training strategy leads to learning the spatio-temporal relationships better, such that when minimal data is available, the model can learn faster. Based on these results, we used the 2D spatial SD model and temporal attention layers pre-trained on our internal dataset as the starting checkpoints for training. For a fair comparison, the baselines SurVGen and SparseCtrl were finetuned with these pre-trained checkpoints. We followed the same training recipe for our proposed SurgiFlowVid model.

### D.3. SurgiFlowVid training

Based on these results we opted to use the temporal layers trained on our internal dataset as the pre-trained model. This offers the advantage that, the SurgiFlowVid training time reduces and also we can avoid the over-fitting of the dataset given the fact that there exists only limited training data from the downstream datasets. We fine-tune the pre-trained temporal attention layers using our proposed dual-prediction U-net module. The optical flow frames are extracted using the RAFT model (Teed & Deng, 2020). For SurgiFlowVid training, we extract clips of 16 frames at a frame rate of 5 for all the datasets. The hyperparameter details are mentioned in Tab. 23.

### D.4. Downstream model training

For the action recognition task (SAR-RARP50), we used the MviT-v2 (Li et al., 2022) model from the SlowFast library[5]. We downsampled the videos to $224 \times 384$ pixels for training with a temporal resolution of 5 fps. Image augmentations such as PCA jitter, RGB scale shift, brightness and contrast shift, random flipping with scale cropping was used along with inverse frequency balancing during the training on the real data. For additional details on the model, readers can refer to SlowFast repo. We followed the similar recipe for the GynSurg dataset. The model were trained for 150 epochs with a learning rate of $1e^4$ with the best model being chosen using a validation dataset.

For the GraSP dataset, we used the similar settings from the TAPIS model[6]. It is to be noted that we do not compare the values directly to the work from (Ayobi et al., 2025) on the GraSP dataset. This is due to the fact that the results reported from the TAPIS model have been obtained directly using the test set as the selection criteria during training. We create a separate validation set from the training set which we use as the selection criteria of the trained model. The test set is clearly separated during the training of both diffusion and downstream models to avoid any data leakage. For the combined training of real and synthetic videos, we opted for a simple and easier strategy than rejection sampling as proposed in (Venkatesh et al., 2025a). We sampled a batch of data points such that 25% of this batch consists of synthetic videos while the remaining 75% includes the real data samples. We chose this method as it works on the fly during training and the time and effort in rejecting synthetic samples are drastically reduced.

For the surgical tool presence detection task, we used the Swin transformer (Liu et al., 2021) base model. The videos were resized to a resolution of $384 \times 384$ during training with augmentations such as RGB channel shift, scaled cropping and temporal shift. We trained the models using binary cross entropy loss with weighted sampling to include the imbalance in the surgical tools. Finally, for laparoscope motion recognition, we utilize a ResNet3D (Hara et al., 2017) model to classify motion categories from input clips using the cross-entropy loss.

## E. Qualitative Results

---

[5]https://github.com/facebookresearch/SlowFast
[6]https://github.com/BCV-Uniandes/GraSP/tree/main/TAPIS

| Hyperparameter | Image fine-tuning | Video-pretraining | SurgiFlowVid training |
|---|---|---|---|
| **Dataset** | | | |
| No. of samples | 4000 | 7000 | Train split of the dataset |
| Resolution | $512 \times 512$ | $256 \times 256$ & $512 \times 512$ | $512 \times 512$ |
| Video length | - | 16 frames | 16 frames |
| Sample rate | - | 5 | 4-5 |
| Context length | - | 16 | 16 |
| **Model params** | | | |
| Pre-trained model | SDv-1.5 | Pre-trained on internal | Pre-trained on internal |
| Params frozen | - | Spatial layers | Spatial layers |
| **Temporal layers** | | | |
| Depth | - | 2 | 2 |
| Temporal resolution | - | $[1, 2, 4, 8]$ | $[1, 2, 4, 8]$ |
| Head channels | - | 16 | 16 |
| No. of heads | - | 8 | 8 |
| Position encoding | - | sinusoidal | sinusoidal |
| PE dim | - | 24 | 24 |
| Cross attention dim | - | 32 | 32 |
| Act.function | - | GeLU | GeLU |
| **Training params** | | | |
| Optimizer | AdamW | AdamW | AdamW |
| Learning rate | $1e^{-4}$ | $1e^{-5}$ | $1e^{-5}$ |
| Lr warm steps | 500 | 5000 | 5000 |
| Lr scheduler | cosine | cosine | cosine |
| $\beta_1$ | 0.9 | 0.9 | 0.94 |
| $\beta_2$ | 0.999 | 0.999 | 0.995 |
| Weight decay $\omega$ | $1e^{-2}$ | - | - |
| Train steps | 3000 | 125k | $75 - 125$k |
| **Train timestep** | | | |
| Diffusion step | 1000 | 1000 | 1000 |
| Noise schedule | linear | linear | linear |
| $\beta_0$ | $1e-4$ | 0.00085 | 0.00085 |
| $\beta_T$ | 0.02 | 0.012 | 0.012 |
| **Sampling params** | | | |
| Sampler | DDPM | DDIM | DDIM |
| Steps | - | 50 | 50 (30 for SVE) |
| CFG scale | 6.5 | 5.5 | 5.0 |
| **Device requirements** | | | |
| GPU-type | A100-40GB | H200-80GB | H200-140GB |
| No. of gpus | 1 | 1 | 1 |

*Table 23.* Hyperparameters for training the 2D and the temporal attention layers of the diffusion model. SVE denotes *Sparse visual encoder* used for conditional generation.

## **Action:** Pull the suture

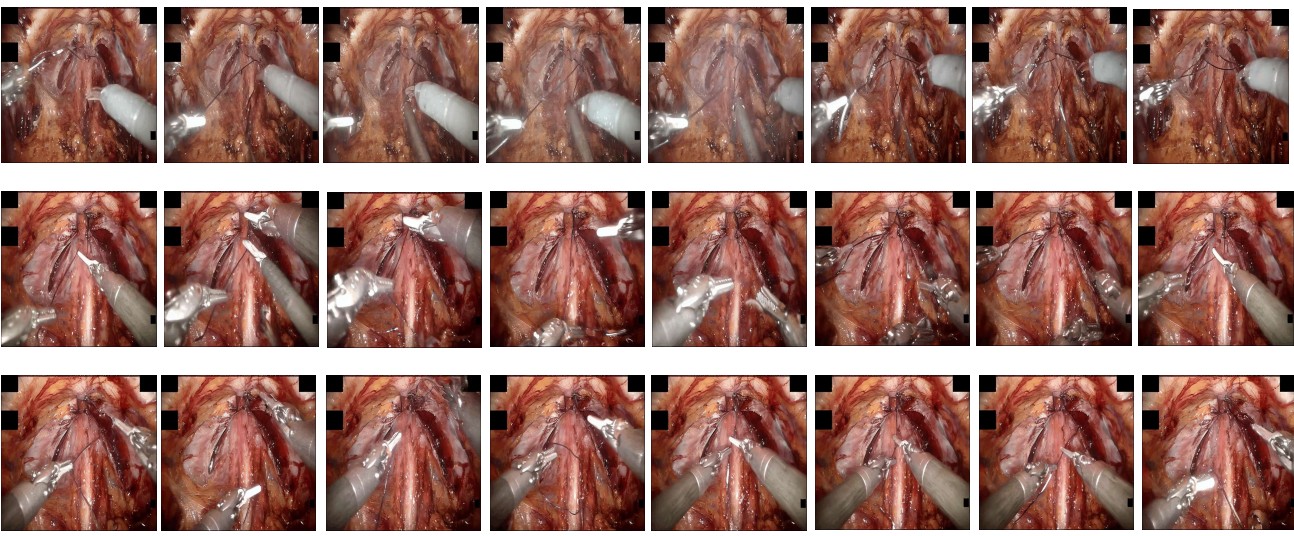

*Figure 11.* Results from SurgiFlowVid with text conditioning on GraSP dataset.

## **Action:** Tie the suture

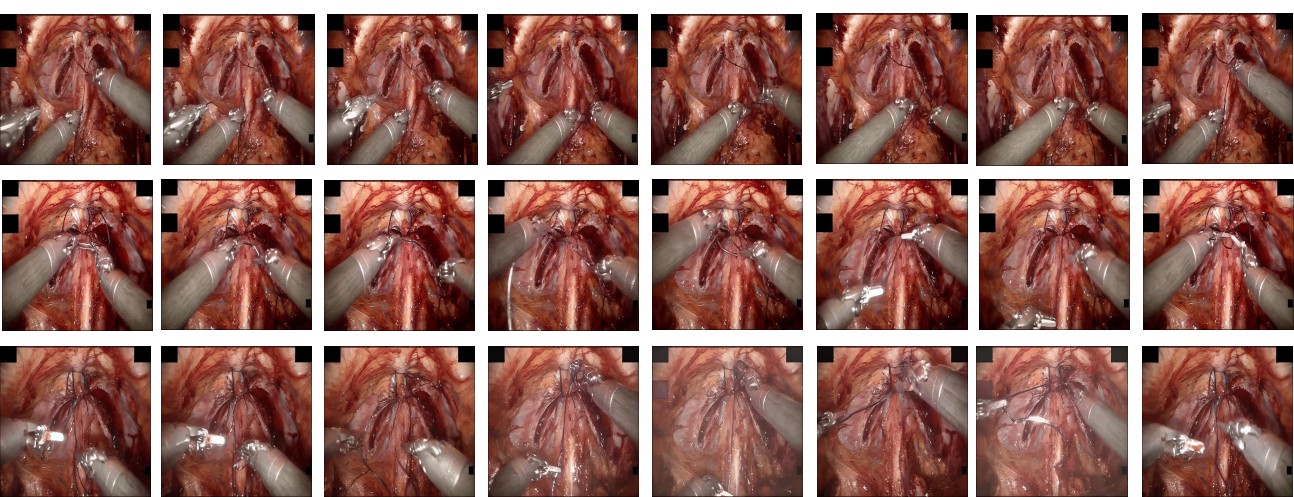

*Figure 12.* Results from SurgiFlowVid with text conditioning on GraSP dataset.

# Action: Cut the suture or tissue

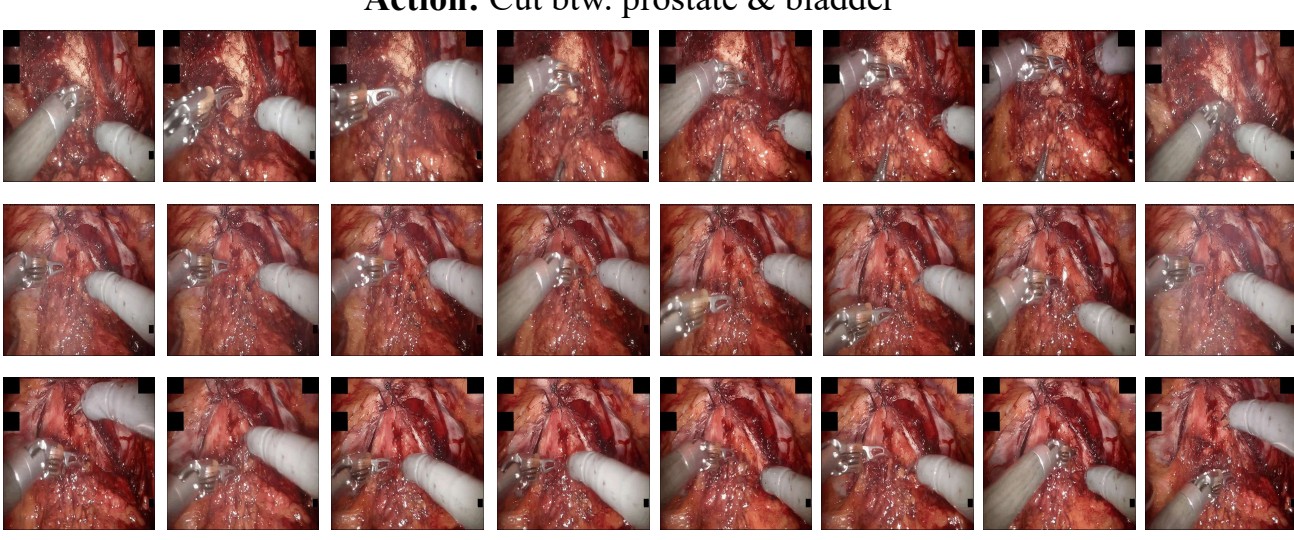

*Figure 13.* Results from SurgiFlowVid with text conditioning on GraSP dataset. In the 2nd row, we notice the presence of smoke as the tissue is cauterized using the tool.

# Action: Cut btw. prostate & bladder

*Figure 14.* Results from SurgiFlowVid with text conditioning on GraSP dataset.

*RGB conditioning frames*

Generated frames

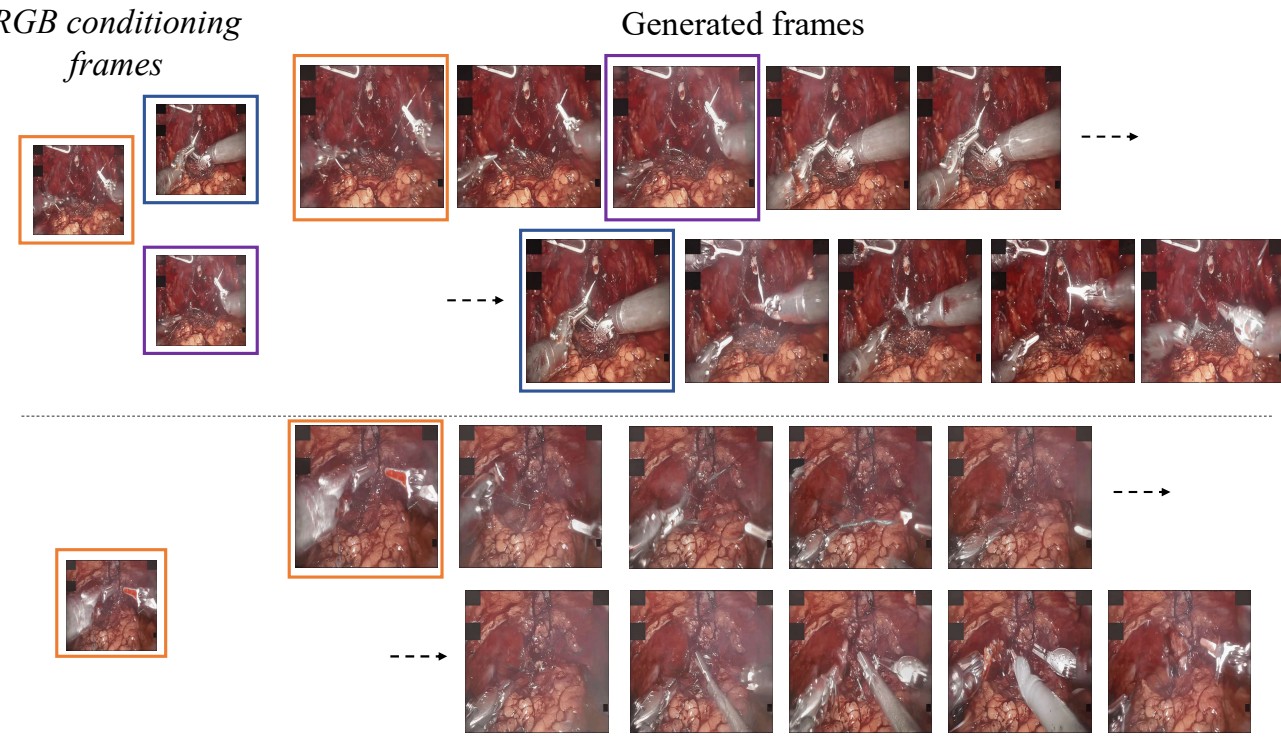

*Figure 15.* Results from SurgiFlowVid with RGB conditioning on GraSP dataset.The frames on the left indicate the sparse conditioning frames and the left frames indicate the generated video frames. The coloured boxes show the position of the corresponding condition frame. The dotted arrow indicates the next subsequent frames. The action corresponds to *pull the suture*.

*RGB conditioning frames*

Generated frames

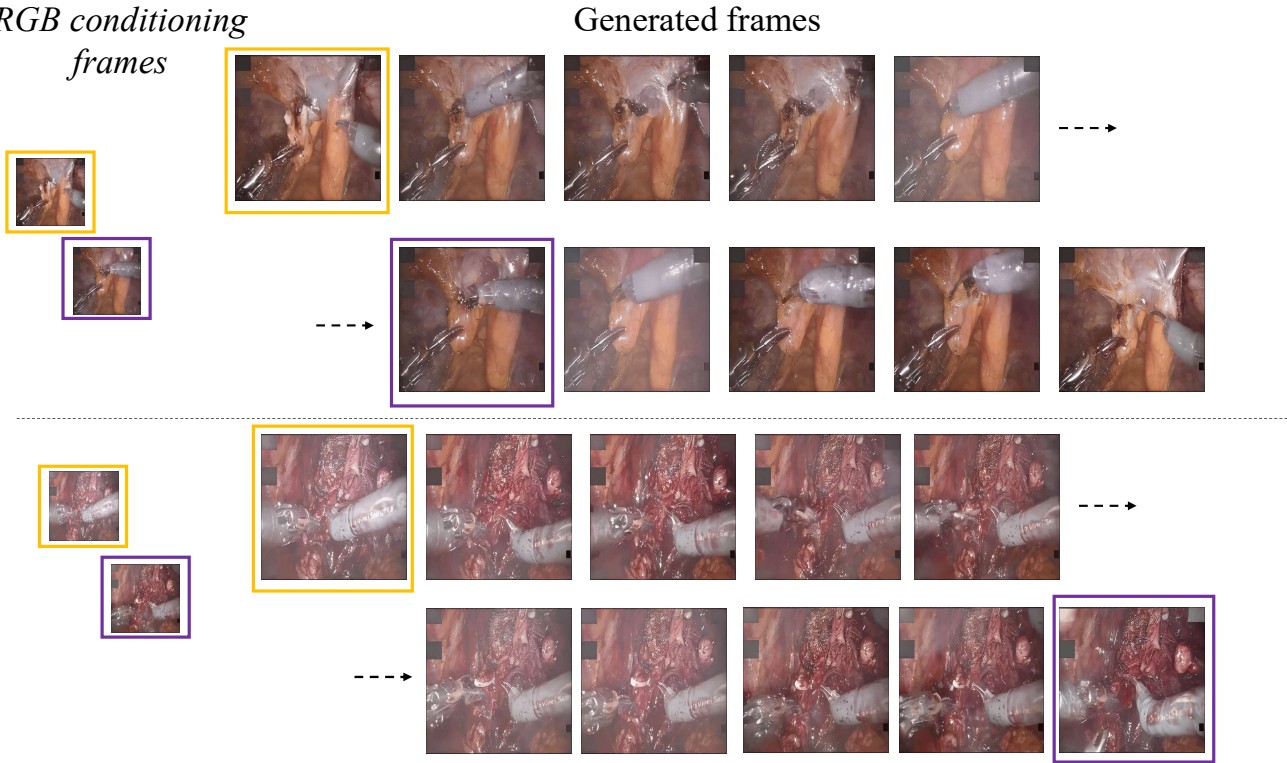

*Figure 16.* Results from SurgiFlowVid with RGB frame conditioning on GraSP dataset. The action corresponds to *cut the tissue*.

