# OpenReview forum: "Mitigating Surgical Data Imbalance with Dual-Prediction Video Diffusion Model"
_ICML.cc/2026/Conference — ICML 2026 regular_

### Official Review · Reviewer_Lix2 · 2026-02-13

**Soundness:** 2
**Presentation:** 3
**Significance:** 3
**Originality:** 2
**Overall Recommendation:** 4
**Confidence:** 3

**Summary:**

The paper proposes SurgiFlowVid, a diffusion-based framework designed to mitigate data imbalance in surgical video datasets. The method extends an existing surgical video diffusion framework (SurV-Gen) by introducing a dual-prediction module that jointly denoises RGB frames and optical flow maps, and a sparse visual encoder for conditional generation using sparse segmentation masks or RGB frames. Synthetic samples generated by the model are used to augment under-represented classes, leading to improvements in several downstream tasks across three surgical datasets.

**Compliance With Llm Reviewing Policy:**

Affirmed.

**Final Justification:**

I believe that future work could further explore leveraging more advanced vision foundation models to address medical tasks. At the same time, I encourage the authors to go beyond simply combining existing components and instead develop more task-specific technical contributions tailored to the unique challenges in the medical domain.

**Key Questions For Authors:**

see weaknesses

**Limitations:**

yes

**Strengths And Weaknesses:**

**Strengths**:
1. The problem of data imbalance in surgical video datasets is well-motivated and important.
2. The evaluation is quite comprehensive, where the authors evaluate on three datasets and multiple downstream tasks  (action recognition, tool detection, motion prediction). The ablation on synthetic data attributes is particularly well-designed.
3. Using optical flow as a temporal inductive bias is reasonable

**Weakness**:

1. The proposed approach primarily builds upon existing diffusion-based frameworks (e.g., SurV-Gen, Stable Diffusion, and prior video generative techniques) and extends them by incorporating optical flow as an additional supervisory signal. While this integration is well-motivated and empirically effective, the resulting contribution appears to be more of an engineering extension rather than a substantial methodological advancement.
2. Video diffusion models have advanced rapidly in recent years. Modern video-native architectures are designed with inherent spatio-temporal modeling capabilities and no longer rely on extending image diffusion models with additional temporal layers to capture motion dependencies. For instance, recent video diffusion backbones (such as Wan2.1-1.3B) demonstrate substantially stronger generation quality and motion coherence without such extensions. To improve the technical novelty of the work, the authors could consider how to integrate their proposed strategy into more modern video diffusion architectures, rather than building upon comparatively outdated paradigms.
3. Although LPIPS and CLIP-based metrics are reported, they mainly capture perceptual fidelity and semantic alignment. Since the proposed method claims improvements in motion modeling, it would be more appropriate to include quantitative evaluations that directly measure spatio-temporal coherence. Could the authors clarify whether such experiments were conducted?

---

> ### Author Rebuttal · Authors · 2026-03-30
>
> We thank the reviewer for their insightful feedback. We are glad that the reviewer found our problem formulation, proposed framework important and well motivated, and appreciate the positive remarks on our extensive evaluation.  We address the raised questions below,
>
> **1. Technical contribution**
>
> We position our work as a *principled, application-driven contribution* tailored for the unique constraints of surgical healthcare data. To our knowledge, this is the first framework to concurrently integrate: (i) RGB-optical flow dual-prediction, (ii) sparse, annotation-light controllable conditioning, and (iii) empirical validation across diverse tasks including action recognition, tool detection, and motion prediction for surgical science. The consistent performance gains across multiple datasets demonstrate that this is not merely an incremental change, but a robust solution to the critical challenge of surgical data imbalance (mentioned as strengths by you and all the other reviewers). We believe this level of integrated innovation is highly aligned with the venue’s focus on application-driven ML for healthcare use cases.
>
> **2. Adaptability to larger and newer architectures**
>
> Our choice of a ≈450M parameter model was motivated by the computational constraints. We conducted **additional experiments** using *Wan2.1-1.3B as a backbone* for text-only generation. The results show our dual-prediction approach SurgiFlowVid continues to outperform SurV-Gen (with rejection sampling) across both balanced (A2, A3) and underrepresented (A1,A5) classes.
>
> | Training data | A1 | A2 | A3 | A5 |
> |:---:|:---:|:---:|:---:|:---:|
> | SurV-Gen (RS) | 0.42 | 0.68 | 0.80 | 0.17 |
> | SurgiFlowVid | 0.48 | 0.67 | 0.82 | 0.24 |
> ------
> We modified initial layers to accommodate concatenated RGB and optical flow signals, enabling joint denoising. These results indicate that the dual-prediction approach is beneficial in improving spatio-temporal coherence even when integrated into modern, billion-scale video architectures. Due to the constraints of the rebuttal timeline (as it takes 1.7 mins to generate a video using Wan2.1), we included two of the represented and underrepresented classes from the SAR-RARP50 dataset.
>
> **3. Measuring spatio-temporal coherence**
>
> We agree that LPIPS and CLIP scores are imperfect proxies for motion quality. Additionally, we evaluate intra-set *motion diversity* using V-JEPA embeddings (Appendix B.8), providing a more direct measure of motion coherence beyond pixel-level metrics. We argue that downstream task performance is in fact the most meaningful measure of spatio-temporal coherence for this work (mentioned by reviewer sXE1). This is directly evidenced in our synthetic data attributes analysis (Table 1), where systematic degradation of spatial structure (noise, elastic deformations) or temporal structure (frame shuffling) consistently reduces downstream performance thereby confirming that the downstream models are sensitive to both.

---

> > ### Author Rebuttal · Reviewer_Lix2 · 2026-04-03
> >
> > Thanks for your response. This is a relatively complete piece of work, and based on this, I am willing to raise my score.
> >
> > I believe that future work could further explore leveraging more advanced vision foundation models to address medical tasks. At the same time, I encourage the authors to go beyond simply combining existing components and instead develop more task-specific technical contributions tailored to the unique challenges in the medical domain. Although the rebuttal is intended to highlight the strengths of the work, I hope the authors will continue to pursue genuine technical contributions.

---

> > > ### Author Response · Authors · 2026-04-06
> > >
> > > We thank the reviewer for their insightful feedback and for the score update. We are pleased that our rebuttal addressed your questions regarding the completeness of this work. We find your suggestions for more task-specific technical contributions very motivating. We fully intend to pursue these directions in our future research to better address the unique data challenges in the medical domain.

---

### Official Review · Reviewer_sXE1 · 2026-03-04

**Soundness:** 3
**Presentation:** 3
**Significance:** 3
**Originality:** 2
**Overall Recommendation:** 5
**Confidence:** 3

**Summary:**

The paper proposes SurgiFlowVid, a video diffusion framework designed to mitigate class imbalance in surgical video datasets by generating synthetic clips of under-represented actions. The model introduces a dual-prediction diffusion module that jointly predicts RGB frames and optical flow during training to improve temporal motion modeling, and a sparse visual encoder that enables conditioning on sparse signals such as segmentation masks or RGB frames. Synthetic videos are added to the real training data and evaluated through downstream tasks including surgical action recognition, tool presence detection, and laparoscope motion prediction on three surgical datasets. Results show consistent improvements when synthetic data from the proposed method is used.

**Compliance With Llm Reviewing Policy:**

Affirmed.

**Final Justification:**

I believe this paper meets the bar for acceptance for icml conference

**Key Questions For Authors:**

1) The experiments appear restricted to a relatively small set of surgical procedures. Could the authors clarify the scope and diversity of the training data and how flexible the model is with synthetic data?

2) Improvements are mainly demonstrated through downstream task performance, while direct evaluation of generation quality is limited. Could the authors suggest additional ways to evaluate the quality of generated videos (e.g., perceptual metrics or expert assessment)? Having said that, the practical usefulness of the synthetic data is clearly demonstrated and way more important in my opinion, so please treat this question more as an open question.

**Limitations:**

yes

**Strengths And Weaknesses:**

**Soundness.**

The overall approach is technically reasonable and well motivated. The idea of incorporating optical flow as an auxiliary prediction signal during diffusion training provides a plausible inductive bias for temporal consistency. Synthetic data can be generated in a controllable way which increases its usefulness. The experimental setup is very solid, with evaluation on three datasets and **multiple downstream tasks** - very important to motivate new data generation. The paper also includes ablations and comparisons with recent surgical video generation baselines. Supplementary material includes additional extensive, in-depth analysis.

**Presentation.**

The paper is generally clearly written and well structured. The method description is understandable, and the experimental section is comprehensive. The narrative connecting synthetic video generation with downstream improvements is easy to follow.

**Significance.**

The work addresses an important problem in surgical data science, where datasets are often highly imbalanced and rare events are difficult to model. Generating synthetic surgical videos for rare classes is a practically relevant direction, and the downstream evaluation demonstrates that such synthetic data can improve task performance.


**Originality.**

The method is well engineered, however the **novelty is somewhat limited** -**I find this its main weakness**. The approach largely builds on existing diffusion-based video generation frameworks (quite old Stable Diffusion), combining temporal diffusion models with optical-flow supervision and sparse conditioning. Contribution mainly lies in engineering work, application to surgical video generation and solid experimental section (incl. downstream tasks). I'd like to emphasise that for me this is **a good enough reason to accept** the paper at ICML conference.

---

> ### Author Rebuttal · Authors · 2026-03-30
>
> We thank the reviewer for their feedback and positive remarks on our approach, the significance of the problem and the extensive evaluation conducted. We address the raised comments as follows:
>
> **1. Scope and diversity of the training data was questioned.**
>
> In this work, we focused on *publicly available datasets* with action-level annotations, covering two robotic prostatectomy datasets (SAR-RARP50, GraSP), one laparoscopic hysterectomy dataset (AutoLaparo), and one gynecological dataset (GynSurg). These datasets span meaningfully different surgical contexts like *robotic vs. laparoscopic* settings, different anatomical regions, and varying tool sets while the results indicate consistent gains observed across all four datasets suggesting good generalizability.
>
> Regarding flexibility, our framework requires only a small set of video clips from under-represented classes with no dense annotations. Optical flow is computed automatically via RAFT at negligible cost compared to manual annotations. The sparse visual encoder supports RGB frames or segmentation masks adapting to varying annotation budgets. These properties make SurgiFlowVid broadly applicable to new procedures with minimal adaptation effort.
>
> **2. Additional metrics for evaluation**
>
> We thank the reviewer for the balanced perspective and agree that downstream task performance is the most meaningful measure in our setting. Beyond this, we report CLIP and LPIPS scores (Appendix B.7) for quantifying the *perceptual quality* and include the intra-set *motion diversity* using V-JEPA embeddings (Appendix B.8). As an open direction for future work, we can employ off-the-shelf object tracking models as an additional measure to quantify the spatio-temporal coherence (tools and anatomies) of the generated videos.

---

> > ### Author Rebuttal · Reviewer_sXE1 · 2026-04-03
> >
> > thank you for answering my questions, I am maintaining my accept score.

---

> > > ### Author Response · Authors · 2026-04-06
> > >
> > > We would like to thank the reviewer again for their valuable feedback. We especially value your perspective on the importance of downstream evaluation. We appreciate your time in evaluating our rebuttal and are grateful for the positive assessment of our work.

---

### Official Review · Reviewer_9RQM · 2026-03-06

**Soundness:** 4
**Presentation:** 3
**Significance:** 3
**Originality:** 4
**Overall Recommendation:** 5
**Confidence:** 3

**Summary:**

This paper introduces SurgiFlowVid, a video diffusion framework that generates synthetic surgical videos for under-represented classes via joint RGB-optical flow denoising and sparse visual conditioning to alleviate data imbalance.

Experiments on three datasets show 10–20% gains across surgical action recognition, tool detection, and laparoscope motion prediction compared to baselines.

**Compliance With Llm Reviewing Policy:**

Affirmed.

**Final Justification:**

The authors have solved my problems.

**Key Questions For Authors:**

The identified weaknesses, including limited temporal adaptability to long-sequence surgical scenarios, insufficient validation of synthetic data’s unique value compared to strong augmentation methods, ambiguous definition of under-represented classes, inadequate inter-frame continuity in sparse frame injection and limited technical innovation in core architecture, require comprehensive addressing for me to consider increasing the score.

**Limitations:**

See weakness.

**Strengths And Weaknesses:**

Strengths:

1. SurgiFlowVid targets a critical practical issue in surgical data science, class imbalance, with a solution that avoids dense annotations and aligns with real healthcare data constraints.

2. The dual-prediction and sparse conditioning design enhances spatio-temporal consistency of synthetic videos, and the framework demonstrates strong generalizability across tasks and datasets.

Weakness:

1. The model’s temporal adaptability is constrained, as it is only designed for short-sequence generation (4 seconds/16 frames) and cannot accommodate long-sequence surgical scenarios such as surgical phase recognition. The feasibility of extending it to autoregressive generation remains unvalidated, which limits its utility to fine-grained tasks.


2. The necessity of synthetic data lacks sufficient validation. The work fails to conduct in-depth comparisons with strong data augmentation techniques to demonstrate the unique value of synthetic data, and only proves effectiveness through simple methods like data duplication or frame shuffling.

3.The definition and classification of under-represented classes seems ambiguous. Although the imbalance ratio is referenced, there is no clear rationale for dataset-specific IR thresholds nor sensitivity analysis on these thresholds, resulting in poor generalizability of the classification logic.

4. The inter-frame continuity of sparse frame injection is inadequately guaranteed. When guiding generation with 3–5 sparse frames, tool position drift persists despite the use of cross-attention for spatial information propagation, indicating incomplete technical mechanisms for ensuring spatio-temporal consistency.

5. The dual-branch design and sparse conditional guidance are adaptations of existing methods, without breakthrough architectural innovations tailored to surgical scenarios. The core innovation lies in application integration rather than the development of novel methodologies.

---

> ### Author Rebuttal · Authors · 2026-03-30
>
> We thank the reviewer for their constructive feedback and for acknowledging the practical significance of our work for surgical science and the generalization of our approach across tasks and datasets. We address the raised concerns below:
>
> **1. Adaptability to longer sequence scenarios**
>
> The focus of this work was on *fine grained surgical tasks* in contrast to long-context phase recognition tasks. These short surgical actions provide information on a finer granularity presenting a more localized and in-depth analysis of the surgical context [1].  We already acknowledge the limitation to short clip generation explicitly in Section 5. However, all evaluated downstream tasks such as action recognition, tool presence detection, and laparoscope motion prediction are clip-level tasks where short segments are the natural unit of analysis. Furthermore, addressing class imbalance in this context does not inherently require long sequences. Extending to longer sequences via autoregressive generation is a promising future direction (combining with Self-forcing [2]) of our work.
>
> **2. Clarification on the necessity of synthetic data**
>
> We would like to clarify a misunderstanding. The experiments in Table 1 are ablations identifying key attributes synthetic data must satisfy and not comparisons against augmentation strategies. For all downstream experiments, the real training data already incorporates strong augmentations (Appendix D.4) and inverse frequency balancing (weighting factor for the classes). Synthetic data is added on top of this strong baseline, meaning reported gains reflect the *unique value of synthetic data* beyond augmentation alone. Additionally, Appendix B.10 provides dedicated comparisons against video-level augmentation baselines, where SurgiFlowVid consistently outperforms these approaches.
>
> **3. Definition of under-represented classes**
>
> The choice of imbalance ratio (IR) is inherently dataset and application dependent. Importantly, our method is **threshold-agnostic**, SurgiFlowVid generates synthetic samples for any user-defined set of classes without framework modifications. The consistent gains across four datasets with varying imbalance distributions support broad applicability. We will add a clearer definition to the revised manuscript.
>
> **4. Inter-frame consistency under sparse conditioning**
>
> We acknowledge tool position drift under sparse conditioning as a genuine limitation, explicitly discussed in Section 5 and Figure 5. However, this is a fundamental challenge in any sparse-to-dense generation framework. Our analysis in Appendix B.9 indicates SurgiFlowVid shows better tool position overlap than baseline (SparseCtrl), and in Figure 3, where competing methods exhibit more severe spatial inconsistencies. The robustness of our spatio-temporal modeling is evidenced by consistent performance gains in the surgical tool presence detection task, where our approach outperforms baselines across multiple tool categories.
>
> **5.  Technical innovation**
>
> This concern closely mirrors a point raised by the reviewer Lix2 (Q1.) and we refer to our response for a detailed discussion. Briefly, we position our work as an *application-driven contribution* that is explicitly valued by this venue. To our knowledge, addressing class imbalance with generative modeling in surgical context is an important and meaningful contribution to the field (acknowledged as strengths by you and all the other reviewers). As this is a particularly nascent area of research, we view SurgiFlowVid as an important first step toward resolving the data bottleneck in surgical AI.
>
> References:
>
> 1. Ayobi et al., Pixel-Wise Recognition for Holistic Surgical Scene Understanding, 2024
>
> 2. Huang et al., Self Forcing: Bridging the Train-Test Gap in Autoregressive Video Diffusion, 2025

---

> > ### Author Rebuttal · Reviewer_9RQM · 2026-04-02
> >
> > Thanks for response, I will raise the score.

---

> > > ### Author Response · Authors · 2026-04-06
> > >
> > > We thank the reviewer once again for their constructive feedback and for the time dedicated to evaluating our rebuttal. We are glad to have addressed the concerns raised and truly appreciate the positive update to the score.

---

### Official Review · Reviewer_cbcn · 2026-03-12

**Soundness:** 3
**Presentation:** 3
**Significance:** 2
**Originality:** 2
**Overall Recommendation:** 4
**Confidence:** 3

**Summary:**

This paper proposes SurgiFlowVid, a diffusion-based framework for generating synthetic surgical videos to mitigate class imbalance in surgical datasets. The core contributions are: (1) a dual-prediction U-Net that jointly denoises RGB frames and optical flow maps to improve temporal coherence; and (2) a sparse visual encoder that conditions generation on sparse segmentation masks or RGB frames, removing the need for dense annotations. The framework is evaluated on three surgical datasets across action recognition, tool detection, and laparoscope motion prediction, reporting consistent gains of 10–20% over baselines.

**Compliance With Llm Reviewing Policy:**

Affirmed.

**Final Justification:**

My concerns have been adequately addressed by the author's responses. I raise my score.

**Key Questions For Authors:**

The performance improvements depend substantially on the proprietary ~7,000-video pre-training set. Could the authors provide an ablation comparing SurgiFlowVid trained without this internal pre-training against the same baseline? This would clarify whether dual-prediction or pre-training data is the dominant factor.

In Table 2, SurgiFlowVid without the SVE module (text-only, λp>0) already outperforms SurV-Gen without rejection sampling. Is the gain attributable purely to the optical flow loss, or does the higher output resolution (512×512 vs. 256×256) also play a role? A resolution-controlled ablation would strengthen this claim.

**Limitations:**

The authors discuss limitations adequately, including the restriction to short-context tasks, occasional tool position drift with sparse conditioning, and directions for future work.

**Strengths And Weaknesses:**

The paper targets a genuine bottleneck in surgical AI — severe class imbalance with annotation scarcity — and proposes a solution that avoids the costly dense-annotation requirements of prior work. The sparse conditioning design is pragmatic and well-motivated for real clinical settings.

A non-trivial portion of the performance gain likely stems from pre-training on an internal ~7,000-video surgical dataset, which is unavailable to the community. Baselines are also fine-tuned from these checkpoints, which partially mitigates but does not eliminate the reproducibility concern. The contribution of this pre-training versus the dual-prediction module itself is not cleanly disentangled.

The method builds directly on SurV-Gen and SparseCtrl, with the main additions being the optical flow auxiliary loss and the integration of the sparse encoder. While the combination is effective, the individual components (flow-guided generation, sparse conditioning) are well-established. The paper would benefit from a more explicit theoretical or analytical justification for why optical flow specifically—rather than other temporal signals—is the right inductive bias in this low-data regime.

---

> ### Author Rebuttal · Authors · 2026-03-30
>
> We thank the reviewer for their feedback and are encouraged that our work tackles an important bottleneck in surgical AI in a pragmatic, clinically relevant way. We address the questions raised below.
>
> **1. Influence of pre-training on performance**
>
> Regarding the pre-training strategy, we clarify that our curated in-house dataset primarily provides *reduced training time* rather than being the fundamental driver of performance. To decouple the dual-prediction module’s influence from the pre-training data, we conducted **additional ablation study** on the SAR-RARP50 dataset with no pre-training. We did not notice a large difference in performance, while the number of training steps needed increased 2x for training from scratch.
>
> | Train data | Pre-training | Num. train steps | A1 | A2 | A3 | A4 | A5 | A6 | Mean |
> |:---:|:---:|:---:|:---:|:---:|:---:|:---:|:---:|:---:|:---:|
> | SurgiFlowVid | $\checkmark$ | 90K | $0.43\pm_{0.12}$ | $0.65\pm_{0.07}$ | $0.77\pm_{0.07}$ | $0.63\pm_{0.11}$ | $0.11\pm_{0.03}$ | $0.35\pm_{0.12}$ | $0.49\pm_{0.04}$ |
> | SurgiFlowVid | $\times$ | 170K | $0.42\pm_{0.10}$ | $0.66\pm_{0.06}$ | $0.77\pm_{0.03}$ | $0.64\pm_{0.12}$ | $0.10\pm_{0.02}$ | $0.36\pm_{0.11}$ | $0.49\pm_{0.06}$ |
> -------------------------
>
> Training the model from from scratch required 170k steps to converge, nearly **double the training steps** to achieve a comparable score to the pre-trained model. These results show that the dual-prediction module is the primary source of improvement. We ensure fair comparison by using these pre-trained checkpoints as a consistent starting point for all baselines. While pre-training accelerates the learning of spatio-temporal features [1,2]  it is not a prerequisite for achieving the reported gains. We will include this ablation in the revised version of the paper.
>
>
> **2. Resolution-controlled ablation study**
>
> To address potential resolution confounding, we conducted **additional controlled ablation** matching the 256×256 resolution of SurV-Gen. At matched resolution, SurgiFlowVid (text-only) shows comparable performance to SurV-Gen with a notable gain for the "pick the needle" class suggesting the *optical flow (dual-prediction module) provides meaningful inductive biases* for certain under-represented classes.
>
> | Training data | Resolution | A1 | A2 | A3 | A4 | A5 | A6 | Mean |
> |---|---|---|---|---|---|---|---|---|
> | SurV-Gen | 256×256 | $0.31\pm_{0.19}$ | $0.64\pm_{0.07}$ | $0.77\pm_{0.06}$ | $0.60\pm_{0.10}$ | $0.13\pm_{0.10}$ | $0.37\pm_{0.18}$ | $0.46\pm_{0.03}$ |
> | SurgiFlowVid | 256×256 | $0.40\pm_{0.09}$ | $0.63\pm_{0.04}$ | $0.77\pm_{0.06}$ | $0.61\pm_{0.08}$ | $0.10\pm_{0.04}$ | $0.34\pm_{0.11}$ | $0.47\pm_{0.02}$ |
> | SurgiFlowVid | 512×512 | $0.43\pm_{0.12}$ | $0.65\pm_{0.07}$ | $0.77\pm_{0.07}$ | $0.63\pm_{0.11}$ | $0.11\pm_{0.03}$ | $0.35\pm_{0.12}$ | $0.49\pm_{0.04}$ |
>
> As a note, for downstream model training, we resize all the videos to the same resolution. The modest overall difference at 256×256 indicates that both the optical flow and higher resolution jointly contribute to the gains in Table 2 (text-only). The dual-prediction module independently improves performance, while higher resolution offers complementary benefits by preserving fine-grained surgical details such as needles and thread. We appreciate the reviewer's observation in identifying this and will incorporate these results into the revised manuscript.
>
> References:
> 1. Blattmann et al., Align your Latents: High-Resolution Video Synthesis with Latent Diffusion Models, 2023
>
> 2. Blattmann et al., Stable Video Diffusion: Scaling Latent Video Diffusion Models to Large Datasets, 2023

---

### Decision · Program_Chairs · 2026-04-30

**Decision:**

Accept (regular)

**Comment:**

Reviewers appreciate the paper’s focus on a practically important problem in surgical AI: using generative models to address severe class imbalance under limited annotations. The final reviewer consensus is positive, with two Weak Accepts and two Accepts after rebuttal. While reviewers note that the method builds on existing diffusion-based components and that the architectural novelty is moderate, they also agree that applying generative modeling to this problem in surgical video understanding is worthwhile and supported by solid downstream results across multiple datasets and tasks. The rebuttal addresses the main concerns, including the role of proprietary pre-training, the potential impact of resolution, the transferability to stronger backbones, and the evaluation of temporal consistency. The AC therefore supports acceptance.